# The bZIP transcription factor BIP1 of the rice blast fungus is essential for infection and regulates a specific set of appressorium genes

Karine Lambou[1,2☯]*, Andrew Tag[3☯], Alexandre Lassagne[2], Jérôme Collemare[1,4], Pierre-Henri Clergeot[1,5], Crystel Barbisan[6], Philippe Perret[6,7], Didier Tharreau[2,8], Joelle Millazo[2,8], Elia Chartier[2], Ronald P. De Vries[9], Judith Hirsch[2,10], Jean-Benoit Morel[2], Roland Beffa[6], Thomas Kroj[2], Terry Thomas[3], Marc-Henri Lebrun[1,11]

**1** CNRS-Bayer Crop Science, UMR 5240 MAP, Lyon, France, **2** Plant Health Institute of Montpellier (PHIM), Montpellier University, INRAE, CIRAD, Institut Agro, IRD, Montpellier, France, **3** Department of Biology, Texas A&M University. College Station, Texas, United States of America, **4** Westerdijk Fungal Biodiversity Institute, Utrecht, The Netherlands, **5** ASP Bourgogne Franche-Comté, Dijon, France, **6** Biochemistry Department, Bayer Crop Science SAS, Lyon, France, **7** Bayer S.A.S. Crop Science Division Global Toxicology- Sophia Antipolis Cedex, France, **8** Plant Health Institute of Montpellier (PHIM), CIRAD, Montpellier, France, **9** Fungal Physiology, Westerdijk Fungal Biodiversity Institute & Fungal Molecular Physiology, Utrecht University, Utrecht, The Netherlands, **10** Pathologie Végétale, INRAE, Montfavet, France, **11** Université Paris-Saclay, INRAE, UR 1290 BIOGER, Palaiseau, France

☯ These authors contributed equally to this work.
* karine.lambou@umontpellier.fr

**Data Availability Statement:** Transcriptomic data from this study have been submitted to the NCBI - under accession no. GSE18823.

## Abstract

The rice blast fungus *Magnaporthe oryzae* differentiates specialized cells called appressoria that are required for fungal penetration into host leaves. In this study, we identified the novel basic leucine zipper (bZIP) transcription factor BIP1 (B-ZIP Involved in Pathogenesis-1) that is essential for pathogenicity. BIP1 is required for the infection of plant leaves, even if they are wounded, but not for appressorium-mediated penetration of artificial cellophane membranes. This phenotype suggests that BIP1 is not implicated in the differentiation of the penetration peg but is necessary for the initial establishment of the fungus within plant cells. *BIP1* expression was restricted to the appressorium by both transcriptional and post-transcriptional control. Genome-wide transcriptome analysis showed that 40 genes were down regulated in a *BIP1* deletion mutant. Most of these genes were specifically expressed in the appressorium. They encode proteins with pathogenesis-related functions such as enzymes involved in secondary metabolism including those encoded by the *ACE1* gene cluster, small secreted proteins such as SLP2, BAS2, BAS3, and AVR-Pi9 effectors, as well as plant cuticle and cell wall degrading enzymes. Interestingly, this *BIP1* network is different from other known infection-related regulatory networks, highlighting the complexity of gene expression control during plant-fungal interactions. Promoters of *BIP1*-regulated genes shared a GCN4/bZIP-binding DNA motif (TGACTC) binding *in vitro* to BIP1. Mutation of this motif in the promoter of *MGG_08381.7* from the *ACE1* gene cluster abolished its appressorium-specific expression, showing that BIP1 behaves as a transcriptional activator. In summary, our findings demonstrate that BIP1 is critical for the expression of early invasion-related genes in appressoria. These genes are likely needed for biotrophic invasion of the first infected

**Funding:** The author(s) received no specific funding for this work.

**Competing interests:** The authors have declared that no competing interests exist.

host cell, but not for the penetration process itself. Through these mechanisms, the blast fungus strategically anticipates the host plant environment and responses during appressorium-mediated penetration.

## Author summary

The identification of gene regulatory networks controlling pathogenicity is a major research goal for understanding plant infection and for developing new strategies for disease control. Rice is the staple food for half the world's population, but its cultivation is threatened by the rice blast fungus *Magnaporthe oryzae* that causes severe yield losses. This fungus can breach intact plant leaves using specialized cells called appressoria. Here, we have identified in a pathogenicity mutant screen using random insertional mutagenesis, the novel *M. oryzae* bZIP transcription factor BIP1 that is essential for the infection. BIP1 is not implicated in the development of appressoria or the subsequent penetration of host leaves, but is necessary for the initial establishment of the fungus within plant cells. BIP1 orchestrates the expression of a unique set of early invasion-related genes within appressoria, encoding secreted effectors, enzymes, secondary metabolism-related enzymes, and signaling membrane receptors. Our experimental data suggest that *BIP1* controls their expression by interacting directly with a TGACTC motif present in their promoters. Remarkably distinct from other known pathogenicity networks, the BIP1 regulatory network underscores the intricate control of fungal gene expression during infection. BIP1 seems to prepare *M. oryzae* for early biotrophic growth during appressorium-mediated penetration.

## Introduction

The fungus *Magnaporthe oryzae* (syn. *Pyricularia oryzae*) is pathogenic on a wide range of cereals, including wheat, barley, and rice, and is one of the most damaging fungal plant pathogens [1, 2]. In addition, *M. oryzae* has been developed as a model organism for studying fungal-plant interactions [1,3–7]. Its pathogenesis relies on the formation of a specialized cell, the appressorium that mediates penetration into host plant tissues [8, 9]. Appressoria differentiate from hyphal tips of germinating conidia attached to the leaf surface. This process is controlled by chemical and physical cues, such as cuticle monomers, surface hydrophobicity and hardness [6,9]. Initiation of appressorium differentiation requires tight control of the cell cycle as well as autophagic cell death of the conidial cells [10,11]. In addition, it relies on appressorium-related signaling such as the Pmk1 mitogen-activated protein kinase (MAPK) pathway and the cAMP signaling pathway [12]. Appressorium-mediated leaf penetration requires a high internal turgor pressure of up to 8 MPa, built up by the accumulation of glycerol [13]. Melanization of the appressorium cell wall is essential for turgor generation mostly by strengthening its rigidity and preventing solute efflux [13,14]. At the base of the appressorium, an area called the appressorium pore in close contact with the host leaf is devoid of melanin. During the maturation of the appressorium, a penetration peg, is formed at the pore, and breaches the cuticle and plant cell wall of the leaf epidermis using mainly a mechanical force [6]. ROS production by NADPH oxidases as well as septin GTPase-dependent actin re-organization of the cytoskeleton are required for the cell re-polarization at the pore and the formation of the penetration peg [15,16]. The penetrating hypha differentiates bulbous infection

hyphae that colonize the infected host cell in close interaction with the host plasma membrane [17]. At this early stage of the infection, the fungus secretes a wide variety of effectors in the extracellular space located between the fungal cell wall and the plant plasma membrane of which many are translocated into the plant cell [18,19]. These effectors suppress plant immunity and manipulate host cellular processes to facilitate infection [20,21]. Five days after the initiation of the infection, the fungus switches to a necrotrophic lifestyle characterized by the active killing of host tissues, and the differentiation of conidiophores, which results in the formation of necrotic sporulating lesions [5,6].

Regulatory networks specific to each stage of the infection process have been revealed in plant pathogenic fungi [22,23]. Transcription factors (TFs) are a major component of these infection-specific regulatory networks. However, only a small number of infection-related TFs have been identified so far. Their regulatory networks are also mostly poorly defined [22,23]. *M. oryzae* has 495 putative TF-encoding genes [24]. Genome-wide surveys and systematic genetic characterization of different TF families have been performed in *M. oryzae* [24–30]. They identified 24 bZIP TFs, of which 15 have critical functions in pathogenicity, development, stress response, or metabolism [28,29,31–33]. Only a few of the regulatory networks controlled by these bZIP TFs have been investigated. bZIP3 controls the expression of genes involved in the autophagy-dependent turgor build-up in the appressorium [32], MoAP1 controls the expression of a wide range of genes involved in oxidative stress response in the appressorium [31], while MoEITF1 and MoEITF2 control the expression of effectors-coding genes during early stages of infection [33].

In this study, we report the identification of a novel *M. oryzae* transcription factor involved in pathogenicity, *BIP1* (bZIP TF Involved in Pathogenesis-1), whose expression is restricted to the appressorium. *BIP1* deletion mutants were non-pathogenic on rice and barley, and differentiated melanized appressoria unable to penetrate into host leaves but penetrated into artificial cellophane membranes. A genome-wide transcriptional analysis revealed a unique set of genes controlled by BIP1, which defined a novel appressorium-specific regulatory network.

## Results

### The *M. oryzae* insertion mutant M763 is non-pathogenic

An insertional mutagenesis screen was performed in *M. oryzae* isolate P1.2 using restriction enzyme-mediated insertion (REMI, [34]) of a hygromycin resistance cassette. Screening of 1000 hygromycin-resistant REMI mutants for their pathogenicity on detached barley leaves, led to the identification of the non-pathogenic mutant M763. While inoculation of rice and barley leaves with wild type (WT) P1-2 conidia caused disease symptoms 6 days after inoculation (dai), no lesion was observed after inoculation with M763 conidia, even 10–15 dai (**Fig 1A and 1B**). Molecular analysis of M763 identified a single linearized pAN7.1 insertion into the open reading frame (ORF) of the gene *MGG_08118*. A cross between M763 and a wild-type strain of opposite mating type showed that this insertion co-segregated with the non-pathogenic phenotype. 5'RACE experiments identified two alternative transcription start sites (TSS) for *MGG_08118* cDNA. The first TSS localized –234 base pairs (bp) from the ATG start codon and corresponds to the current *MGG_08118* gene model, while a second TSS was localized at -53 bp (**Fig 2**). Sequencing of cDNA clones revealed a 3' UTR of 1217 bp and showed that the gene has 3 introns of 153, 78 and 108 bp (**Fig 2**). To confirm the insertion mutant phenotype, *MGG_08118* was deleted in isolate P1.2 by targeted replacement of its CDS by a hygromycin resistance cassette (**S1 Fig**). *MGG_08118* deletion mutants were non-pathogenic on barley and rice plants as well as on detached barley leaves (**Fig 1**). Complementation of a *MGG_08118* deletion mutant with a *MGG_08118* wild-type allele restored pathogenicity (**Fig 1A and 1B**).

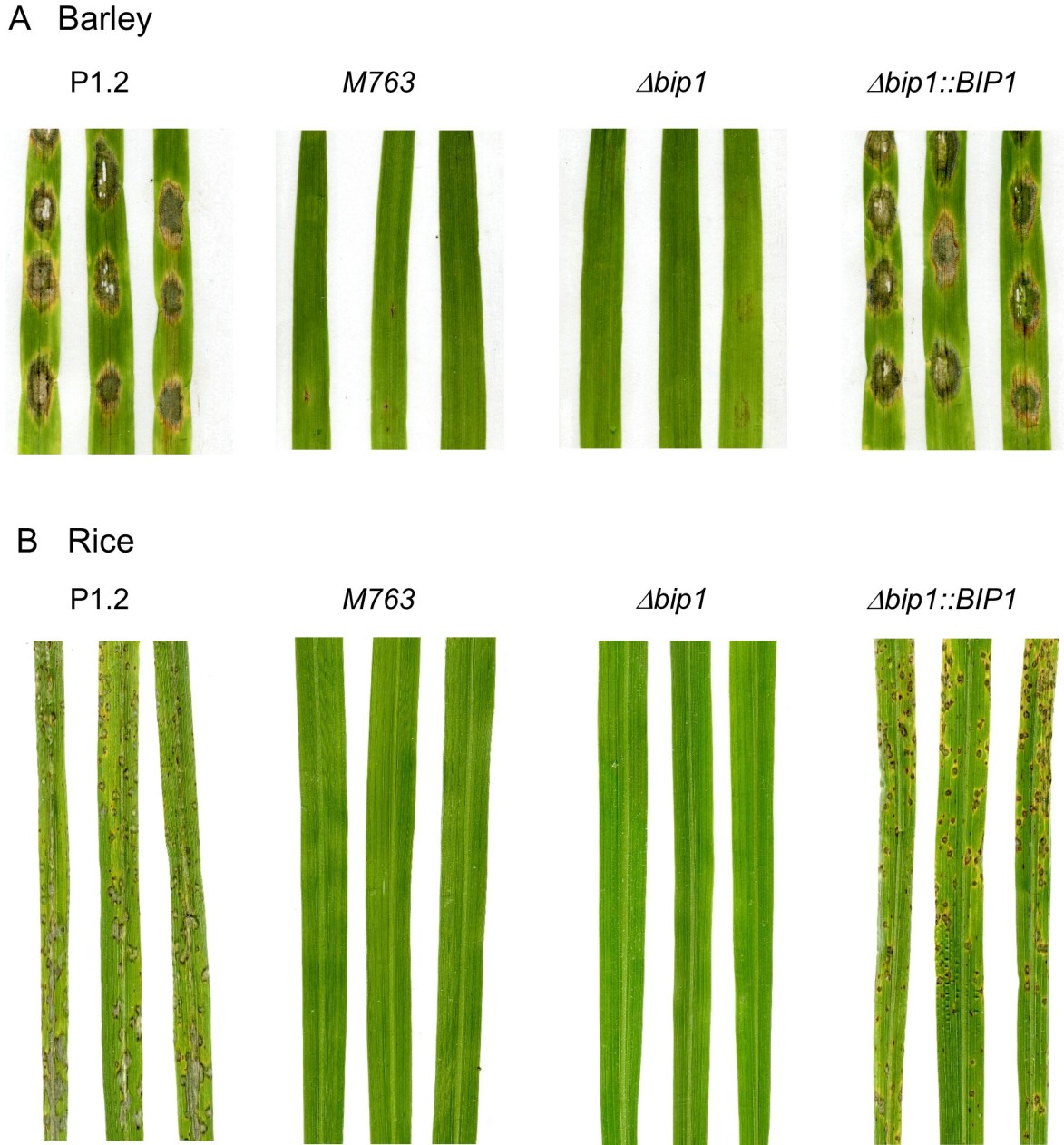

**Fig 1. *M. oryzae* mutants M763 and *Δbip1* were non-pathogenic on barley and rice.** (A) Detached barley leaves were inoculated with droplets of conidial suspensions ($3.10^5$ conidia/mL) from *M. oryzae* wild-type isolate P1.2, insertion mutant M763, a *Δbip1* deletion mutant and *Δbip1* complemented with a wild-type copy of *BIP1 (Δbip1::BIP1)*. Pictures were taken 6 days after inoculation (dai). (B) Four-week-old rice plants were spray-inoculated with conidial suspensions from isolates P1.2, M763 *Δbip1* or *Δbip1::BIP1* ($5.10^4$ conidia/mL) and pictures were taken 7 dai.

The *MGG_08118* deletion mutant, differentiated melanized appressoria that were indistinguishable from WT ones in shape, number and turgor (**Figs 3 and S2**). Therefore, the loss of *MGG_08118* did not impair the differentiation, turgor build-up and maturation of appressoria. *MGG_08118* deletion mutants were also not affected in their conidiation (**S3 Fig**), mycelial growth (**S4 Fig**), or resistance to cell wall and oxidative stress (**S5 Fig**). However, the ability of this mutant to penetrate into barley and rice epidermal cells was completely abolished

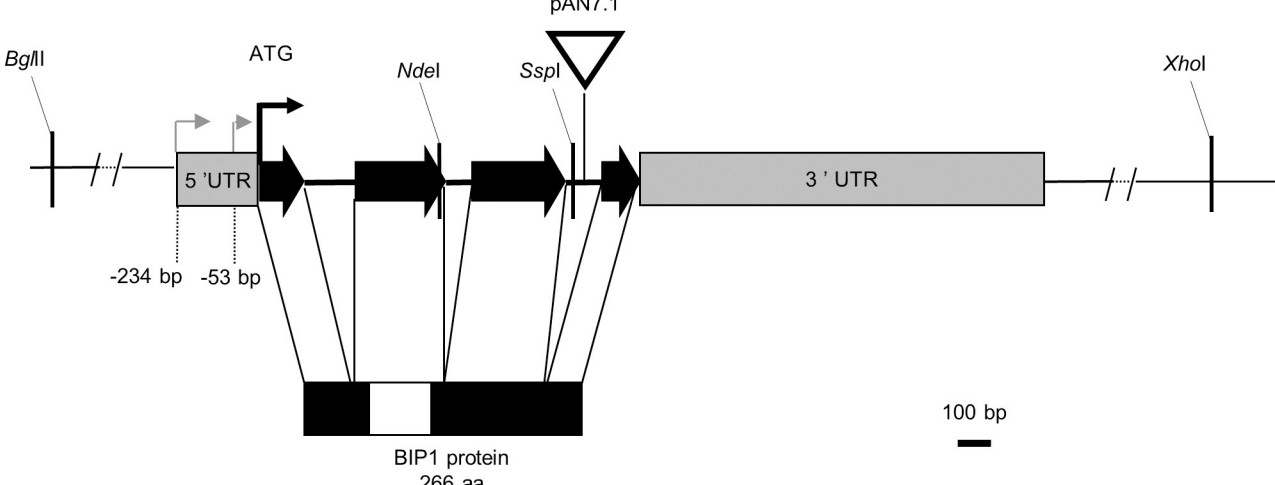

**Fig 2. *M. oryzae* BIP1 locus.** The *BIP1* gene is composed of three introns (black lines) and four exons with a 5'-UTR of 234 bp, a 3'-UTR of 1217 bp (both gray boxes) and a CDS of 798 bp (black arrows). 5'-RACE identified transcription starts at positions –53 bp and– 234 bp from the ATG (grey arrows). In the M763 insertion mutant, plasmid pAN7.1 (white triangle) was inserted in the third *BIP1* intron, downstream of the exon encoding the bZIP domain. Restriction sites used for cloning are displayed. The BIP1 protein (266 aa) has a bZIP domain located near the N-terminal end (white box).

(**Figs 3 and S6**). Indeed, confocal microscopy showed that the mutant was arrested before the development of early infection hyphae, and, presumably, did not produce a penetration peg (**Figs 3 and S6**). The *MGG_08118* deletion mutant was also unable to infect wounded leaves indicating that the formation of invasive infection hyphae was also impaired *in planta* (**S7 Fig**). In contrast, the *MGG_08118* deletion mutant was able to penetrate through appressoria into artificial cellophane membranes. As wild-type and the complemented strain, the *MGG_08118* mutant differentiated bulbous pseudo-infection hyphae within the membrane under the appressoria, forming a star-like network of hyphae within the cellophane (**Fig 4**). These findings indicated that *MGG_08118* is not involved in the differentiation of the penetration peg, or primary infection hyphae. Instead, they suggested that *MGG_08118* is essential for the early establishment of the fungus in its host plant, as evidenced by the absence of infection hyphae in the epidermal plant cell below the appressoria (**Figs 3 and S6**).

## *MGG_08118* encodes a bZIP transcription factor

MGG_08118 encodes a protein of 266 amino acids with one putative nuclear localization signal and a basic leucine zipper (bZIP) domain similar to those of the *Saccharomyces cerevisiae* bZIP TFs GCN4 and YAP1 (**S8A Fig**). The protein domain search tools CD (CD14688), Panther (PTHR11462), and Superfamily (SSF57959), identified a bZIP domain in BIP1 protein sequence (**S8B Fig**), but not Pfam. This domain is composed of a putative DNA interaction domain consisting of a short sequence rich in basic amino acids and four repeats of a leucine zipper motif involved in protein dimerization. Hence, MGG_08118 was named *BIP1* for b̲ZIP TF I̲nvolved in P̲athogenesis-1̲. A BlastP search of the *M. oryzae* 70–15 proteome at NCBI revealed one closely related bZIP TF, MGG_08587, sharing 45% amino acid sequence identity with BIP1. Genes encoding proteins similar to BIP1 and MGG_08587 were identified in Pezizomycotina, but not in Saccharomycotina and Basidiomycota (**Fig 5**). We compared BIP1 and MGG_08587 with the known bZIP TFs of *M. oryzae* and five other Pezizomycotina (*Neurospora crassa*, *Fusarium graminearum*, *Botrytis cinerea*, *Parastagonospora nodorum*, and *Aspergillus nidulans*) by an alignment of their bZIP domain sequences. This alignment was used to

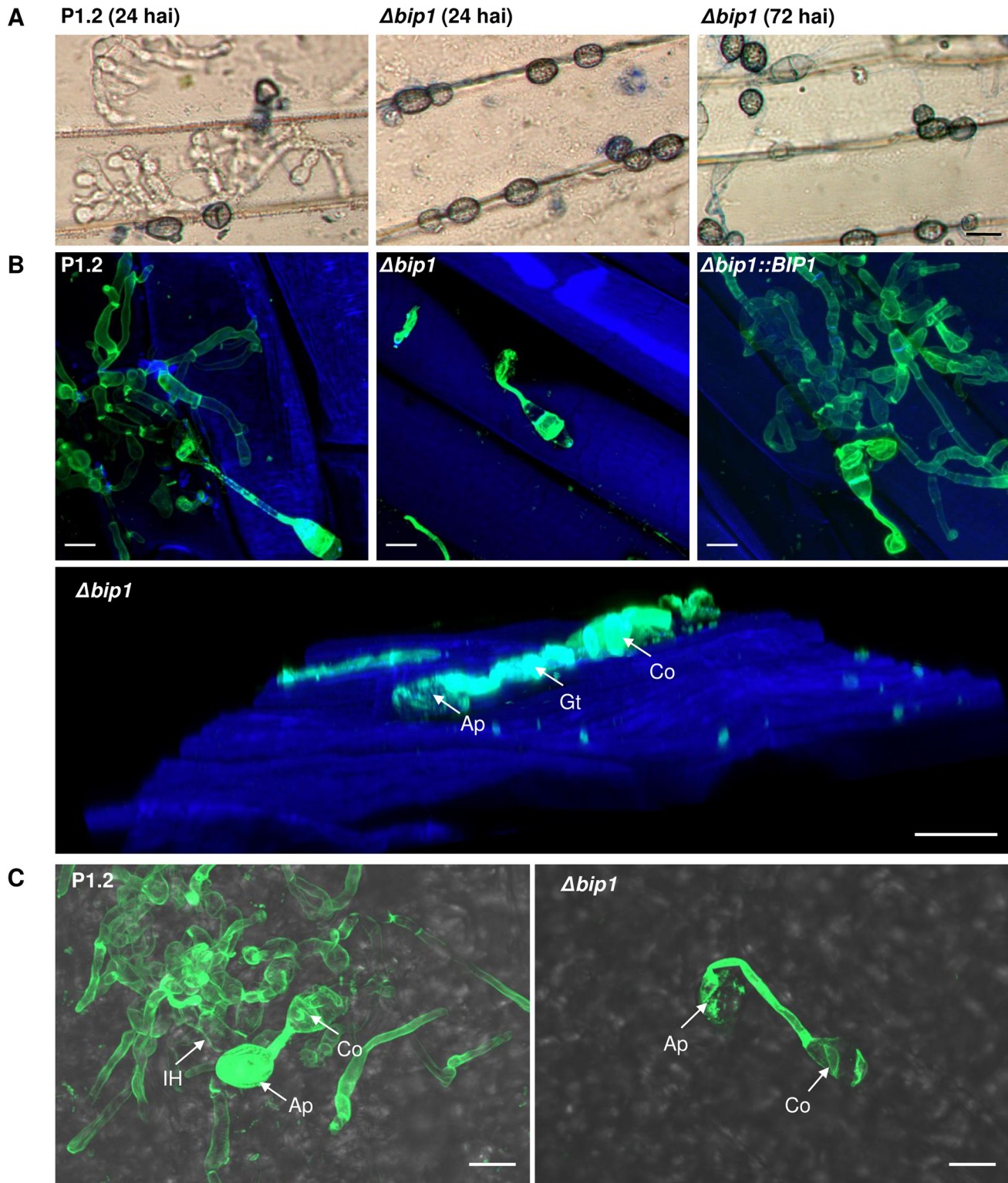

**Fig 3. Loss of BIP1 did not affect appressorium maturation but disabled appressorium-mediated leaf penetration.** (A) Detached barley leaves were inoculated with conidial suspensions of isolates P1.2 (wild-type, WT) or *Δbip1* and peeled 24 and 72 hai for observation under a microscope (x400). Melanized appressoria developed normally in WT and *Δbip1*. Bulbous infection hyphae were visible at 24 hai in WT, but were absent in *Δbip1*, even at 72 hai. Size bar = 10 µm. (B) Observation of hyphal penetration in barley cells at 48 hai using confocal microscopy (63X) and fungal staining with WGA-Alexa488 after tissue fixation. No infection hyphae were observed in epidermal cells of barley leaves infected by *Δbip1* mutant, whereas *Δbip1::BIP1*

and P1.2-infected barley leaves showed numerous invasive infection hyphae resulting from successful penetrations of barley epidermal cells. Fluorescence of WGA-Alexa488-stained fungal cells was excited with 488 nm light and is shown in green. Fluorescence of calcofluor-stained plant cells was excited with 380 nm light and is shown in blue. Lower panel: 3D reconstruction from Z-stack. Size bar = 10 μm. (C) Observation of appressorium-mediated penetration in rice sheath cells of strains P1.2 and Δ*bip1* at 48 hai using confocal microscopy (63X). No infection hyphae were observed in epidermal cells of rice sheaths infected with Δ*bip1* mutant whereas epidermal cells of rice sheaths infected with P1.2 were filled with infection hyphae resulting from penetration events. Fungal cells were stained with the WGA-Alexa488 after tissue fixation. Fluorescence of WGA-Alexa488-stained fungal cells was excited with 488 nm light and is shown in green and combined with a bright-field image of the leave. Individual bright-field and fluorescence images are shown in S6 Fig. Size bar = 10μm. Ap: appressorium, Gt: germ tube, Co: conidium, IH: invasive hyphae.

construct a phylogenetic tree of fungal bZIP TFs (Fig 5). This phylogenetic analysis showed that BIP1 and MGG_08587 are paralogs that cluster with other fungal bZIP TFs as separate clades. BIP1 and MGG_08587 clades are distinct from previously described fungal bZIP clades [28, 29], and they share a common ancestry supported by a bootstrap value of 89% (Fig 5). BIP1 and MGG_08587 clades belong to the bZIP TF superfamily including the FLB, AP1, HAPX, and RSMA clades, but not the superfamily gathering ZIF1, HAC1, ATF1, MEAB, CPC1, IDI4, and METR clades (Fig 5).

## BIP1 is expressed in appressoria and located in the nucleus

Quantitative RT-PCR (qRT-PCR) showed that *BIP1* was expressed at the same high levels in conidia and appressoria differentiated on Teflon and displayed only a low level of expression in mycelium from axenic culture (Fig 6A). In infected barley leaves, *BIP1* was detected as early as 8 hours after inoculation (hai) and reached its maximal level of expression at 17 hai, which coincides with appressorium maturation and precedes host leaf penetration (Fig 6B). To further investigate *BIP1* expression, a Δ*bip1* mutant was transformed with a transcriptional fusion vector carrying three copies of eGFP under the control of the *BIP1* promoter and its 5' and 3'UTR sequences. The fungal transformants were analyzed by fluorescence microscopy at different stages of the infection of barley leaves. GFP fluorescence was detected specifically in mature appressoria with a maximal level of expression between 15 and 48 hai, but not in

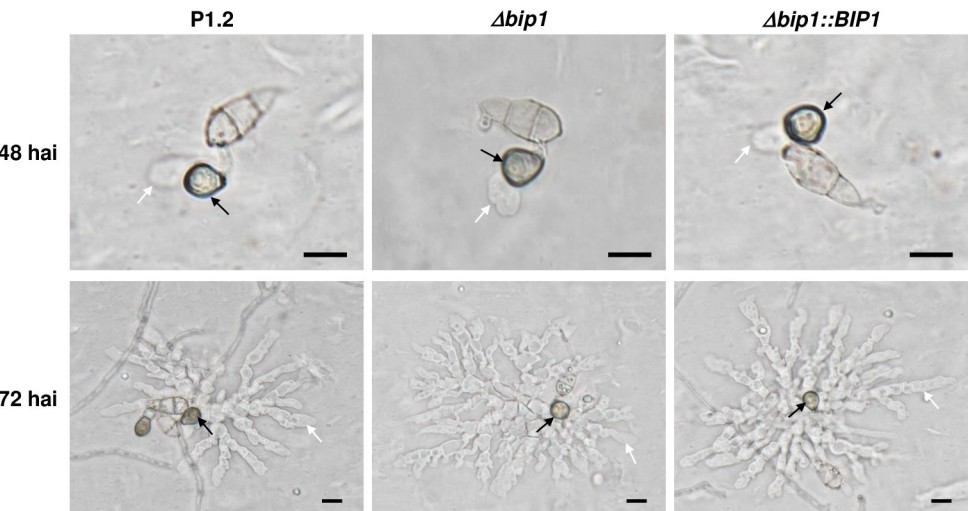

**Fig 4. Appressorium-mediated penetration of Δ*bip1* into cellophane membranes.** Conidia of wild-type (P1.2), Δ*bip1* mutant and Δ*bip1* complemented strains were deposited on cellophane membrane, and appressoria (black arrows) and pseudo-infection hyphae growing into the membrane (white arrows) were visualized by differential interference contrast microscopy at 48 hai and 72 hai. Bar = 10μm.

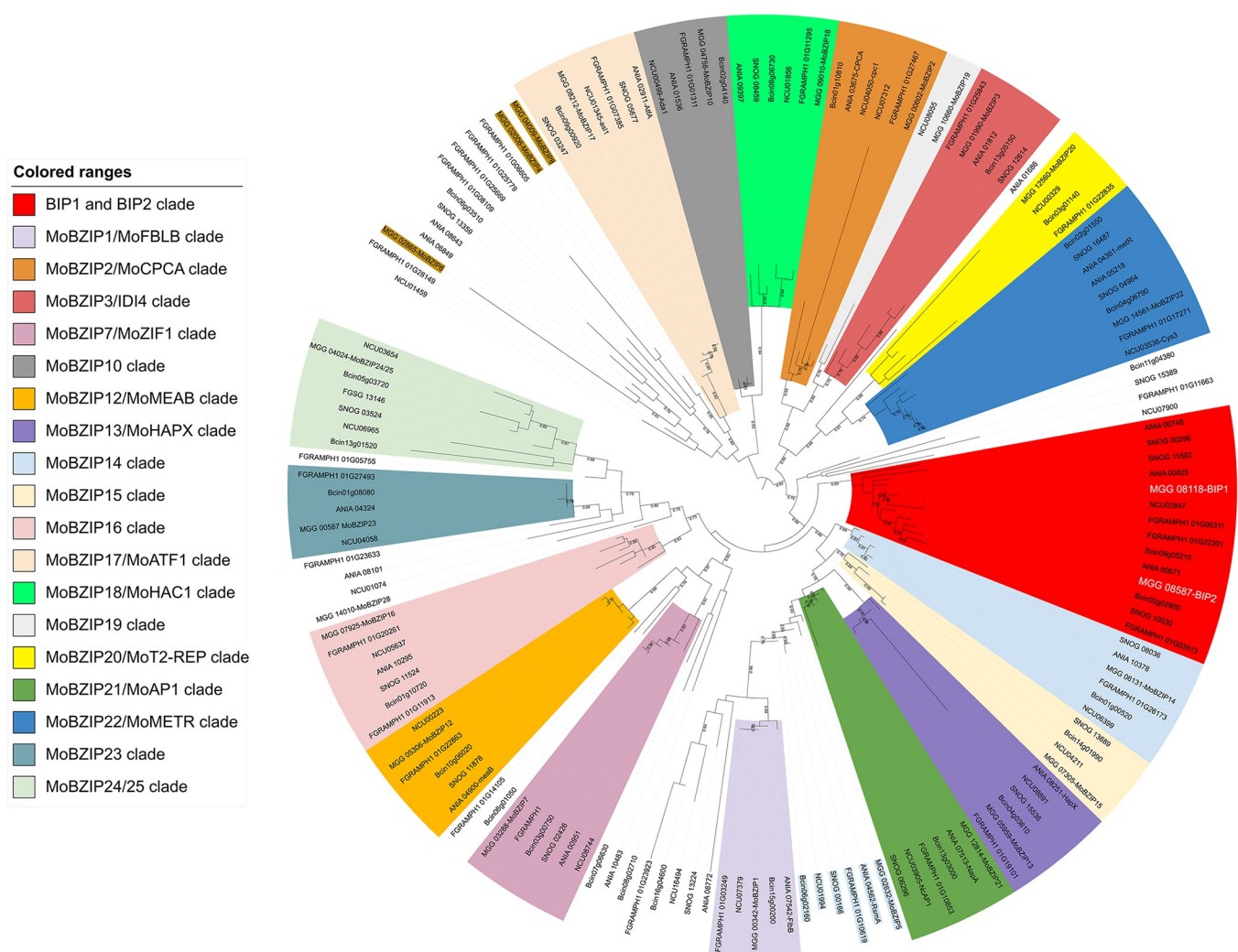

**Fig 5. BIP1 defined a novel clade of bZIP TFs in *Pezizomycotina*.** Phylogenetic tree of bZIP domains of TFs from *M. oryzae* and five selected *Pezizomycotina* species (MGG: *Magnaporthe oryzae*, NCU: *Neurospora crassa*; ANID: *Aspergillus nidulans*, BC1G: *Botrytis cinerea*, SNOG: *Parastagonospora nodorum*, *Fusarium graminearum)*. In red, the clade of BIP1 and its orthologs. bZIP TFs were identified using the Interpro motif IPR004827 and the Superfamily motif SSF57959. In addition, BIP1 and its closest homologs, bZIP TF MGG_08587, as well as their orthologs identified by blastp searches and the Panther motif PTHR11462 were added to this dataset. bZIP domain sequences were extracted from the protein sequences and aligned with ClustaL omega (**S1 data**). The phylogenetic tree was constructed using the maximum likelihood method with RxML.

conidia, mycelium, nor infection hyphae (**Fig 6C**). The absence of GFP fluorescence in conidia despite the detection of *BIP1* mRNA by qRT-PCR, suggested a post-transcriptional control of *BIP1* expression. Such control could involve *BIP1* 5'UTR and 3'UTR sequences present in the transformation construct. To monitor the expression of the BIP1 protein and its subcellular localization, a translational fusion vector carrying a *BIP1-3xeGFP* fusion under the control of the *BIP1* promoter and its 5' and 3'UTRs, was introduced in the *Δbip1* mutant. Confocal laser scanning microscopy detected GFP fluorescence specifically in the nucleus of mature appressoria but not in conidia, nor germinating hyphae and young appressoria (**Fig 6D).** These experiments showed that BIP1 is an appressorium-specific TF with a nuclear localization, expressed after the migration of the nucleus into the appressorium.

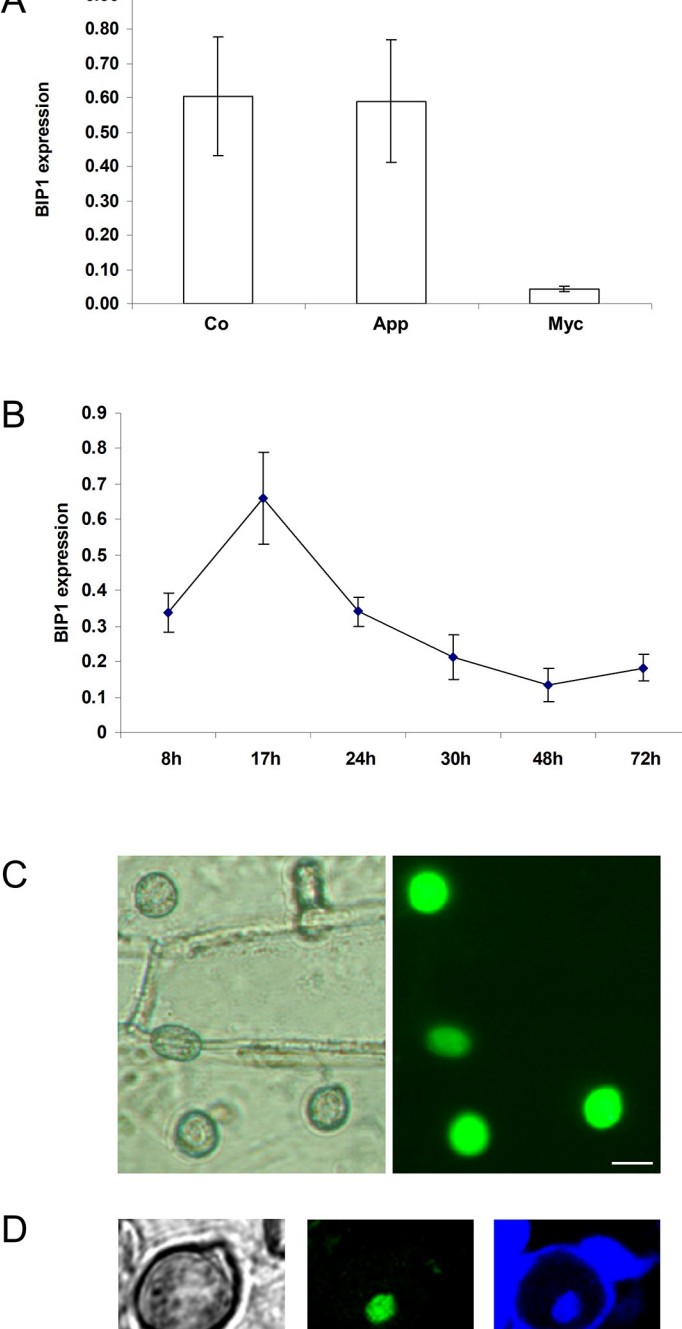

**Fig 6. *BIP1* is specifically expressed in appressoria.** (A) *BIP1* expression in different *M. oryzae* tissues. *BIP1* expression was monitored using qRT-PCR with RNA extracted from conidia (Co), mycelium (Myc), and 24h old appressoria (App) differentiated on Teflon of the P1.2 wild-type strain. (B) *BIP1* expression during infection of barley leaves. Quantification of *BIP1* expression by real-time RT-PCR using RNA from infected barley leaves collected at different time points after droplet inoculation with wild-type conidial suspensions. Data were normalized relative to the constitutively expressed gene ILV5. Each data point is the average of three biological replicates. Standard deviation is indicated by error bars. (C) *BIP1* expression in appressoria during infection. *Δbip1* transformants expressing a *pBIP1::3xeGFP* transcriptional fusion construct were inoculated on barley leaves. Mature appressoria were analysed at 20 hai using transmitted white light (left) or fluorescence microscopy with eGFP filters (right). (D) Localization of BIP1 in nuclei from appressoria. Conidia from a *Δbip1* transformant expressing a *pBIP1::BIP1-3xeGFP* translational

fusion were deposited on a hydrophobic Teflon membrane and appressoria developed 24hai, were stained with DAPI and calcofluor and observed using transmitted light (left), eGFP fluorescence microscopy (middle) and blue fluorescence microscopy (DAPI and calcofluor, right). Scale 10 μm.

## BIP1 is required for the expression of a specific set of genes in the appressorium

To identify genes regulated by BIP1, a genome-wide differential expression analysis of WT and *Δbip1* mutant appressoria differentiated on Teflon membranes was performed using a *M. oryzae* oligonucleotide microarray. This differential expression analysis identified 42 genes down-regulated in *Δbip1* appressoria, but no up-regulated genes (**Table 1**). These BIP1-regulated genes were classified into seven classes according to the cellular functions of their corresponding protein (**Table 1**). The largest class gathered 12 genes coding for proteins involved in secondary metabolism (SM) (**Table 1**). Among them, nine were located within the *ACE1* gene cluster (**Table 1**), a 57 kb genomic region that contains 15 SM genes including the avirulence gene *ACE1* coding for an hybrid polyketide synthase/non-ribosomal peptide synthetase (*PKS-NRPS*) [35, 36]. The second largest class gathered eleven effectors, *i.e.* small secreted proteins (SSPs) without homologies to known protein domains, and included the previously characterized biotrophic effectors BAS2 and BAS3, as well as the SLP2 and AvrPi9 effectors [37–40] *(***Table 1**). The third class was composed of six genes coding for secreted enzymes: two alpha/beta hydrolases, one cutinase, one chitinase, one feruloyl esterase, and one peptidase. The phylogenetic analysis of the feruloyl esterase MGG_03771 showed that this enzyme was related to *A. nidulans* and *A. niger* feruloyl esterases FaeB (**S9 Fig**). Five genes down-regulated in *Δbip1* encoded proteins involved in signaling, mainly G-Protein Coupled Receptors (GPCRs) corresponding to three PTH11-like proteins (MGG_03584, MGG_06535, MGG_02160), and one cAMP receptor-like protein (MGG_10544) [41]. Overall, the BIP1-regulated genes fell into two functional groups, one likely acting on plant cells (SSPs, enzymes, secondary metabolites) and another involved in appressorium signaling pathways.

qRT-PCR experiments were performed with eight BIP1-regulated genes from three functional categories using RNAs from *Δbip1* and wild-type appressoria differentiated on Teflon. All eight genes were down-regulated by at least 2-fold in *Δbip1* appressoria, seven more than 10- fold (**Table 1, qPCR1, *Δbip1*/wt**). Seven of the eight candidate genes were strongly up-regulated in WT appressoria as compared to WT mycelium (**Table 1, qPCR2, Ap/My**). This result was confirmed by independent, published appressorium RNAseq data [42] showing that 93% of the BIP1-regulated genes were up-regulated in mature wild-type appressoria (**Table 1**). Additional qRT-PCR experiments were performed to monitor the regulation of genes from the *ACE1* gene cluster by BIP1 during plant infection. *ACE1* (MGG_12447) and seven of the nine candidate *BIP1* target genes from the *ACE1* gene cluster (**Table 1**) were down-regulated in *Δbip1* at an early stage of barley infection (24 hai) (**S1 Table**). Only two genes, MGG_08386 (BC2) and MGG_15928 (CYP2) were expressed at a higher level in *Δbip1* than in the WT, suggesting that they are regulated differently in appressoria differentiated on Teflon or on plant leaves (**Tables 1** and **S1**). Analysis of the expression of 15 BIP1-regulated genes during initial stages of rice infection (16 hai) showed that the induction of six of them was completely abolished and five others were strongly downregulated (**S2 Table**).

Taken together, our results show that BIP1 coordinates the expression of a specific set of 40 infection-related genes during appressorium-mediated penetration.

**Table 1. Genes down-regulated in _Δbip1_ appressoria differentiated on Teflon.** The expression ratios are log2 transformed. * Comparison of appressoria and _in vitro_ grown mycelium. RNA seq comparing appressoria and germinating conidia from Osés-Ruiz _et al._, 2021 [42]. nid: not in dataset.

| Gene | Class | Function | Gene name | Gene cluster | Reference | μarray _Δbip1_/wt | qPCR1 Ap _Δbip1_/wt | qPCR2 WT Ap/ My | RNAseq WT* Ap/ Gc | EMSA |
|---|---|---|---|---|---|---|---|---|---|---|
| MGG_02420 | Metabolism | Sugar 1,4 lactone oxidase | | | | -1.6 | | | nid | |
| MGG_02530 | Metabolism | Quinate permease | | | | -1.9 | | | nid | |
| MGG_02559 | Metabolism | Molybdenum cofactor sulphurase | | | | -1.6 | | | 2.4 | |
| MGG_03263 | Metabolism | Betaine aldehyde dehydrogenase | BADH2 | | [43] | -1.5 | | | 2.8 | |
| MGG_04240 | Metabolism | FAD oxidoreductase | | | | -1.5 | | | 4.4 | |
| MGG_04738 | Metabolism | Short-chain dehydrogenase | | | | -1.6 | | | 5.1 | |
| MGG_09681 | Metabolism | Gluconolactonase | | | | -2.4 | | | 4.3 | |
| MGG_08236 | Secondary Metabolism | Polyketide synthase | | | | -2.2 | | | 6.3 | |
| MGG_08377 | Secondary Metabolism | O-Methyltransferase | OME1 | ACE1 | [36, 44] | -2.1 | | | 6.6 | |
| MGG_08378 | Secondary Metabolism | Cytochrome P450 | CYP4 | ACE1 | [36, 44] | -2.8 | | | 6.6 | |
| MGG_08379 | Secondary Metabolism | Cytochrome P450 | CYP3 | ACE1 | [36, 44] | -2.4 | | | 7.3 | |
| MGG_08380 | Secondary Metabolism | Enoyl reductase | RAP2 | ACE1 | [36, 44] | -3.8 | -10.0 | 13.3 | 7.4 | Yes |
| MGG_08381 | Secondary Metabolism | Diels-alderase | ORF3 | ACE1 | [36, 44] | -4.1 | -11.3 | 11.1 | 9.4 | Yes |
| MGG_08386 | Secondary Metabolism | Zn finger transcription factor | BC2 | ACE1 | [36, 44] | -1.8 | -5 | 10.8 | 6.6 | No |
| MGG_08387 | Secondary Metabolism | Cytochrome P450 | CYP1 | ACE1 | [36, 44] | -1.8 | | | 7.1 | |
| MGG_08391 | Secondary Metabolism | Enoyl reductase | RAP1 | ACE1 | [36, 44] | -2.7 | | | 7.3 | |
| MGG_11096 | Secondary Metabolism | Thioesterase | | | | -1.2 | | | 4.6 | |
| MGG_13405 | Secondary Metabolism | Terpene synthase | | | | -1.3 | | | 9.7 | |
| MGG_15928 | Secondary Metabolism | Cytochrome P450 | CYP2 | ACE1 | [36, 44] | -1.6 | -3.0 | 11.6 | 10.2 | |
| MGG_02201 | Secreted Enzyme | Peptidase A1 | | | | -4.1 | -9.0 | 11.6 | 8.5 | Yes |
| MGG_03771 | Secreted Enzyme | Feruloyl esterase | FAEB | | [45] | -2.1 | | | 8.2 | |
| MGG_05855 | Secreted Enzyme | α/β Hydrolase | | | | -1.6 | | | 4.1 | |
| MGG_08480 | Secreted Enzyme | α/β Hydrolase | | | | -2.6 | | | 4.5 | |
| MGG_11966 | Secreted Enzyme | Cutinase | | | | -1.6 | | | 7.8 | |
| MGG_17153 | Secreted Enzyme | Chitinase | | | | -1.6 | | | 2.3 | |
| MGG_00751 | Secreted Protein | small secreted protein | | | | -1.1 | | | 1.4 | |
| MGG_03468 | Secreted Protein | Lysm domain protein | SLP2 | | [39] | -2.8 | | | 5.1 | |
| MGG_03504 | Secreted Protein | small secreted protein | | | | -1.6 | | | 1.9 | |
| MGG_05638 | Secreted Protein | small secreted protein | | | | -2 | | | 8.8 | |
| MGG_06666 | Secreted Protein | small secreted protein | | | | -2.3 | | | 2.2 | |
| MGG_07934 | Secreted Protein | small secreted protein | | | | -1.8 | | | 8.0 | |

(_Continued_)

**Table 1.** (Continued)

| Gene | Class | Function | Gene name | Gene cluster | Reference | μarray Δbip1/wt | qPCR1 Ap Δbip1/wt | qPCR2 WT Ap/ My | RNAseq WT* Ap/ Gc | EMSA |
|------|-------|----------|-----------|--------------|-----------|-----------------|-------------------|-----------------|-------------------|------|
| MGG_08428 | Secreted Protein | small secreted protein | | | | -2.8 | | | 7.0 | |
| MGG_09693 | Secreted Protein | Biotrophy assoc.protein 2 | BAS2 | | [38] | -2.2 | | | 2.7 | |
| MGG_11610 | Secreted Protein | Biotrophy assoc.protein 3 | BAS3 | | [38, 40] | -3.1 | | | nid | |
| MGG_12655 | Secreted Protein | small secreted protein | AVR-PI9 | | [37] | -3.2 | | | 6.8 | |
| MGG_17425 | Secreted Protein | small secreted protein | | | | -2.3 | | | 3.6 | |
| MGG_02160 | Signaling | GPCR PTH11 family, CFEM | | | | -1.2 | -1.1 | 3.4 | 2.2 | |
| MGG_03526 | Signaling | N6 Adenine DNA methylase | | | | -1.7 | | | 3.2 | |
| MGG_03584 | Signaling | GPCR PTH11 family, CFEM | | | | -3.3 | -1.7 | 0.5 | 6.4 | Yes |
| MGG_06535 | Signaling | GPCR PTH11 family | | | | -2.9 | -3.9 | 7.2 | 3.8 | Yes |
| MGG_10544 | Signaling | GPCR cAMP Glucose receptor-like | | | | -1.7 | | | 7 | |
| MGG_00545 | Unknown | Unknown | | | | -2.7 | | | 3.7 | |

## BIP1 binds to promoters of genes down-regulated in Δbip1 appressoria

Analysis of promoter sequences 1 kb upstream from the start codon, showed an enrichment of a conserved TGACTC sequence similar to the GCN4-like binding motif in the promoters of the 40 genes down-regulated in Δbip1 appressoria. Indeed, using MEME, the motif was found in 36% of the BIP1-regulated genes and only 12% of the 12.593 promoters of *M. oryzae* (FIMO analysis, p-value $10^{-6}$, [46]). Searching for more degenerated versions of the motif using an in-house script, identified the motif in one or multiple copies in the promoters of all 40 genes.

To analyze whether BIP1 may regulate its target genes by direct binding to these GCN4-like binding motifs, *in vitro* electrophoretic mobility shift assays (EMSA) were performed using recombinant BIP1 protein and radiolabeled 50-bp oligonucleotides centered on the TGACTC sequences of the promoters from six randomly chosen BIP1-regulated genes (S3 Table). BIP1 bound to an oligonucleotide covering the two motifs of the bidirectional promoter shared by *MGG_08380* (*RAP2*) and *MGG_08381* (*ORF3*) from the *ACE1* gene cluster (Fig 7A). This binding was reduced by an excess of non-labeled probes and was strongly reduced or abolished by mutations of, respectively, the first or both core motifs (Figs 7A and 8A). BIP1 also bound to probes centered on the single GCN4-like binding motif of the promoter from *MGG_02201* (peptidase) (Fig 7A). In addition, BIP1 bound to the oligonucleotides covering the GCN4-like motifs of the promoters from the *PTH11-like* genes *MGG_03584*, and *MGG_06535* (Fig 7A). In all cases, BIP1 binding was reduced by competition with unlabeled probes. Only the oligonucleotides corresponding to the four TGACTC motifs of *MGG_08386* coding the BC2 TF from the *ACE1* gene cluster did not bind BIP1 (S3 Table). Taken together, among the five promoters tested with EMSA, four (*MGG_02201; MGG_03584; MGG_06535; MGG_08380/ MGG_08381*) have at least one GCN4-like motif binding to BIP1. The alignment of sequences from oligonucleotides binding to BIP1 *in vitro* highlighted the motif CATGACTCG as a possible extension of the BIP1 binding site sequence (Fig 7B).

## The BIP1 binding site in *MGG_08381* promoter is required for appressorium-specific expression

To test the functional role of the TGACTCG binding site found in the promoters of BIP1-regulated genes, we selected *MGG_08381* from the *ACE1* gene cluster specifically expressed in

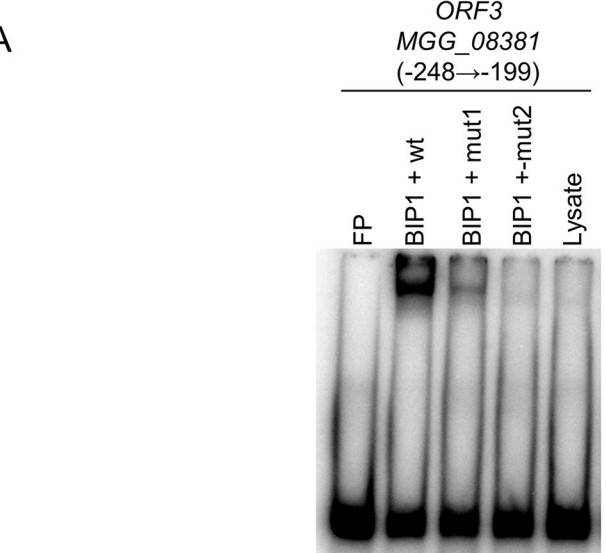

*pORF3*-wt   GCAAAAGGTATTTTC<u>GAGTCA</u>TGCTCC<u>TAGTCA</u>TGGAATAAAAGATGGGA
*pORF3*-mut1 GCAAAAGGTATTTTCAGCGATTGCTCCTAGTCATGGAATAAAAGATGGGA
*pORF3*-mut2 GCAAAAGGTATTTTCAGCGATTGCTCCAGTTACTGGAATAAAAGATGGGA

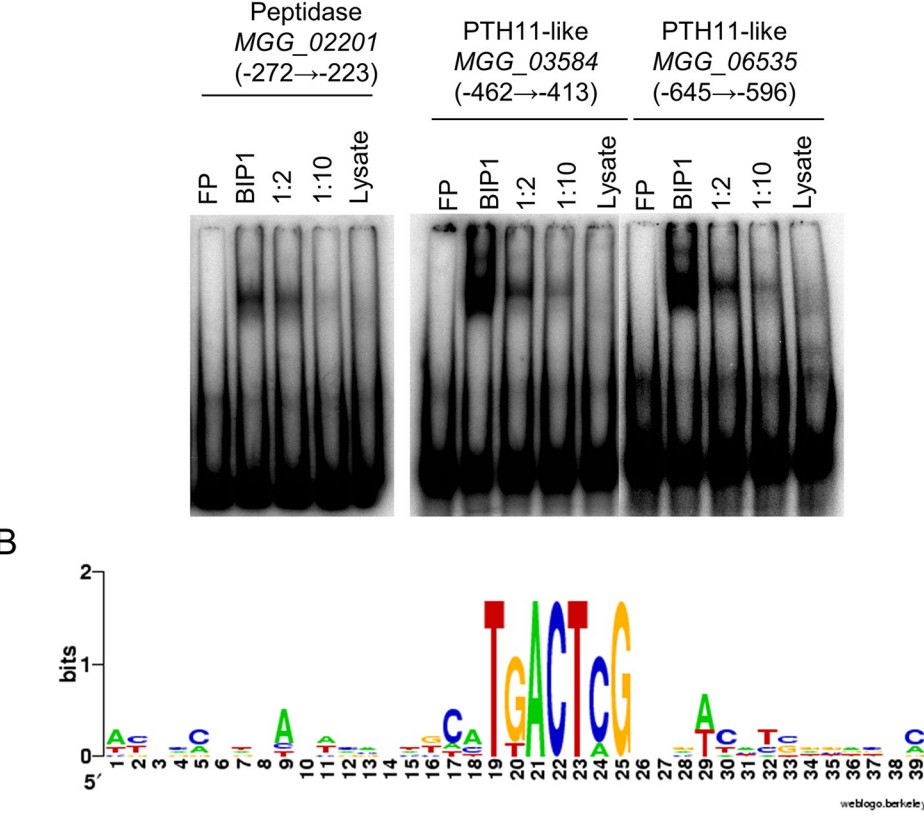

**Fig 7. BIP1 binds to a GCN4-like TGACTC motif.** (A) Radioactively labeled oligonucleotide probes centered on the TGACTC motifs of promoters from MGG_08381/MGG08380 (ORF3/RAP2), MGG_02201 (Peptidase), MGG_03584 (*PTH11*-like), MGG_06535 (*PTH11*-like) genes were incubated in the presence of binding buffer (FP, free probe), BIP1 rabbit reticulocyte lysate (BIP1), BIP1 lysate with unlabeled probe (1:2 or 1:10 molar ratios) or rabbit reticulocyte lysate lacking the BIP1 expression construct (Lysate) before non-denaturing PAGE. For MGG_08381/MGG08380 (ORF3/

RAP2), oligo probes carrying a mutation in the first or both GCN4 motif were tested in addition to the wild-type oligonucleotide. The mutant oligonucleotide probes *pORF3*-mut1 and *pORF3*-mut2 are shown with mutated bases in blue. (B) The oligonucleotides bound by BIP1 in EMSA were aligned on the TGACTC core sequence and submitted to WebLogo to define the BIP1-binding consensus motif.

appressoria [36]. Transgenic *M. oryzae* isolates carrying transcriptional fusions of e*GFP* with either the native or a mutated version of the 378 bp promoter of *MGG_08381* were constructed. In the mutant promoter, the two TGACTCG BIP1 binding motifs as well as the six nucleotides between them were mutated. An oligo probe carrying these mutations did not bind to BIP1 in EMSA (**Fig 8A**). Transformants with the native *MGG_08381.7* promoter displayed a high GFP fluorescence in appressoria during penetration of barley leaves (**Fig 8B**) suggesting that this promoter fragment is sufficient to drive an appressorium-specific expression. On the contrary, transformants carrying the mutated version of the *MGG_08381* promoter did not display eGFP fluorescence in their appressoria (**Fig 8B**). This result shows that the TGACTCG BIP1-binding motifs of *MGG_08381* promoter are required for an appressorium-specific expression.

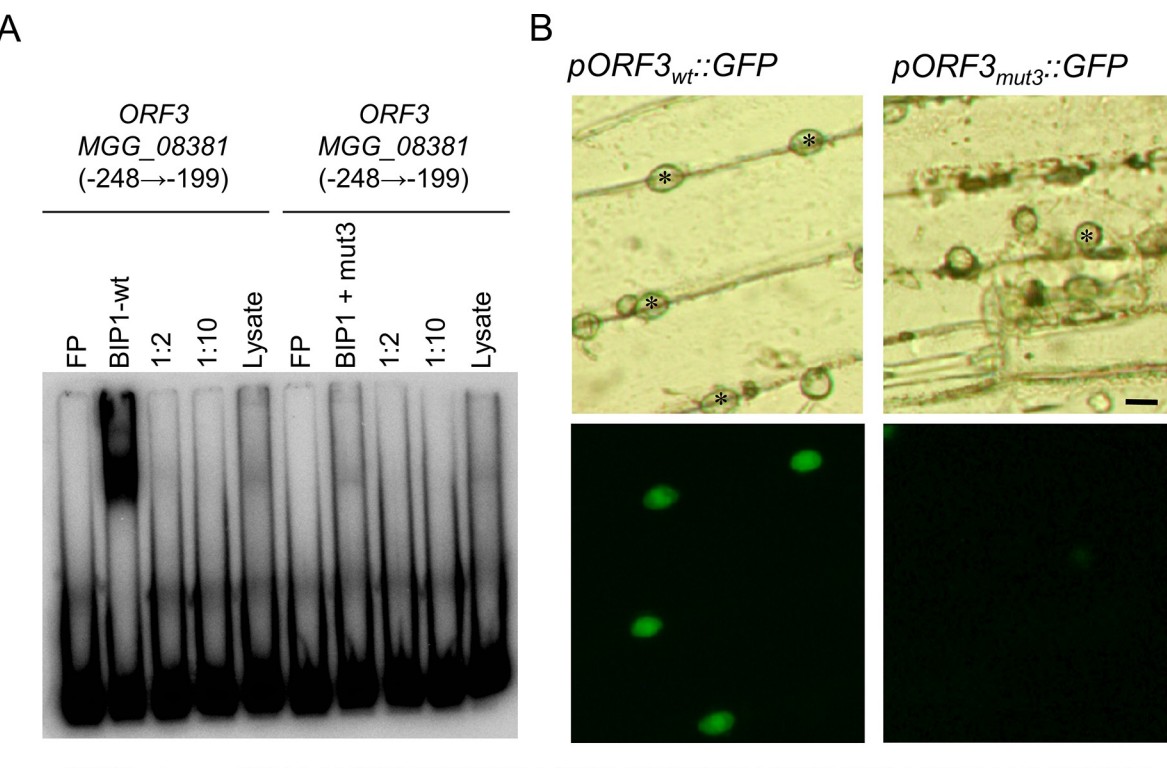

**Fig 8. The BIP1 binding motifs in the promoter of *ORF3* were required for its appressorium-specific expression.** (A) Binding of BIP1 to an *ORF3* promoter fragment relies on the presence of GCN4 motifs. Labeled oligonucleotide probes containing either the two GCN4-like TGACTC motifs of MGG_08381 (ORF3) (left) or a probe mutated for these motifs and the connecting 6 nucleotides (right) were incubated in the presence of binding buffer (FP, free probe), BIP1 rabbit reticulocyte lysate (BIP1), BIP1 lysate with unlabeled probe (1:2 or 1:10 molar ratios), or rabbit reticulocyte lysate lacking the BIP1 expression construct (Lysate) before non-denaturing PAGE. (B) Transgenic *M. oryzae* isolates carrying the GFP reporter gene under the control of the wild-type *ORF3* promoter or a mutant promoter lacking the two BIP1 binding motifs were inoculated on barley leaves and mature appressoria were analyzed 20 hai by fluorescence microscopy for transmitted white light (top) and green fluorescence (bottom). Scale bars = 10 μm.

## Discussion

### BIP1 is a novel *M. oryzae* bZIP transcription factor required for pathogenicity

In this study, we have identified BIP1 (MGG_08118) a novel bZIP TF of *M. oryzae* involved in fungal pathogenicity. BIP1 and its closest paralog (MGG_08587) have not been detected in previous genome-wide surveys of bZIP TFs in the rice blast fungus [28,29]. Consequently, they have never been studied and in particular not in functional analyses using deletion mutants [28,29]. We identified orthologs of *BIP1* and *BIP2* in *Pezizomycotina*, and discovered that they delineate two novel clades of bZIP TFs in this major subdivision of ascomycete fungi (**Fig 5**).

*BIP1* deletion mutants (*Δbip1*) were non-pathogenic on rice and barley leaves (**Fig 1**). Although they differentiated melanized appressoria with normal turgor, they were unable to penetrate into intact or wounded host leaves (**Figs 3, S2 and S7**). This phenotype is different from mutants impaired in appressorium maturation, turgor generation or appressoria adhesion, which cannot penetrate intact leaves, but still infect wounded leaves, such as *Δbuf1*, deficient for melanin biosynthesis, *ΔcpkA*, defective for cAMP signaling and *Δsps1* defective for spermine synthase [47,48].

*M. oryzae* mutants unable to infect wounded plant as *Δbip1*, were affected in genes essential for penetration peg formation like *PLS1*, *NOX2*, *MPS1* or *MST12* [5,15,42,47, 49], or crucial for establishing primary infection hyphae within infected plant cells [50]. Cellophane penetration assays conducted with the *Δbip1* mutant suggested that *BIP1* is not directly involved in the differentiation of the penetration peg. Instead, it is more likely that BIP1 is essential for the early establishment of the fungus within plant cells. Indeed, as the wild type, the *Δbip1* mutant penetrated through appressoria into artificial cellophane membranes, where it formed bulbous pseudo-infection hyphae growing within the membrane under the appressoria (**Fig 4**).

In most other *M. oryzae* mutants with defects in early invasive infectious growth, primary infection hyphae (PIH) are formed in the first infected cell before being arrested [51]. However, a few mutants exhibited a phenotype similar to *Δbip1*, characterized by either the absence or the presence of a very short PIH. Examples of this second type included deletion mutants of the histone deacetylase *TIG1*, the transcription factor *ATF1* or the iron metabolism protein *DES1* [50,52,53]. These three mutants differentiate a few short PIH that are rapidly blocked in the first infected epidermal cell. These mutants displayed an increased sensitivity to $H_2O_2$, which was not observed for *Δbip1*. Their attempted infection triggers strong plant defenses in the first infected cell including increased $H_2O_2$ production, potentially accounting for their arrest. In the case of *Δatf1 and Δdes1*, pathogenicity could be restored by pharmacological inhibition of ROS production, further supporting that non-pathogenic phenotypes are primarily due to an inability to overcome plant defenses. Some *M. oryzae* mutants displayed phenotypes more similar to *Δbip1*. For instance, the deletion mutant of the MIG1 transcription factor from the MPS1/SLT2 signaling pathway was unable to form PIH within rice and onion cells, but like *Δbip1* could penetrate cellophane. Additionally, it invaded heat-killed rice and onion leaf epidermal cells [54]. The deletion mutant of the transcription factor WOR1/GTI1 formed very short PIH within rice and onion cells [55]. Preliminary characterization of its regulatory network suggested differences from that of BIP1 (see below). Based on these examples from the literature, we conclude that BIP1 is essential for early invasive infectious growth. This hypothesis may be further examined in future studies by elucidating its role in establishing within the first infected cell and/or overcoming plant defenses.

Fifteen other bZIP TFs with defective mutant phenotypes, have been described in *M. oryzae* (**S4 Table**). Seven of these bZIP TF mutants display phenotypes associated with either stress response, development (mostly sporulation) or specific growth conditions, but not

pathogenicity on plants (**S4 Table**). Some of them control basic cellular processes, like *Mob-ZIP12*, which encodes an ortholog of *A. nidulans* MeaB, a regulator of nitrogen assimilation [56]. As in *A. nidulans*, *MobZIP12* deletion mutants were unable to grow on synthetic medium containing nitrite or nitrate. However, they were still pathogenic on rice. The other eight *Mob-ZIP* TF deletion mutants were altered in their pathogenicity (**S4 Table**). Six of them were reduced in their pathogenicity (-70 to -80%) and they all displayed additional defects in morphology, development (sporulation, sexual reproduction) and stress responses (osmotic, oxidative). Representatives of this class were *bZIP13*, orthologous to *Aspergillus fumigatus HapX* controlling iron homeostasis [57], bZIP7, an ortholog of *Fusarium graminearum* ZIF1 required for sexual development [58], and *bZIP17*, an ortholog of *A. nidulans* AftA involved in conidia oxidative stress response [59]. The last two MobZIP deletion mutants (*MobZIP21*, *MobZIP22)* were as non-pathogenic as *Δbip1*. The deletion of *MobZIP21*, an ortholog of *N. crassa AP1* involved in the oxidative stress response [60], led to mutants hypersensitive to oxidative stress, highly reduced in sporulation and aerial hyphae formation and differentiating melanized appressoria unable to penetrate into rice leaves [31]. Deletion of *MobZIP22*, an ortholog of *CYS3* from *N. crassa* [61] and *metR* from *A. nidulans* [62] involved in the regulation of sulfur metabolism, resulted in mutants auxotrophic for methionine and cysteine and unable to penetrate rice leaves despite the formation of melanized appressoria. Addition of exogenous methionine rescued the penetration defect of *Δbzip22*. A similar phenotype was described for the *M. oryzae* methionine synthase mutant *Δmet6* [63]. Both *Δbzip22* and *Δmet6* mutants are unable to infect rice leaves, since they are unable to grow on/in infected leaves without methionine supply. Indeed, this amino-acid is not freely available on the surface and in the apoplast of rice leaves. These studies have established that *MobZIP21* and *MobZIP22* are involved respectively in the control of the oxidative stress response and methionine biosynthesis that are basic cellular processes required for infection. On the contrary, the *Δbip1* mutant did not display any other defective phenotypes than a lack of pathogenicity, suggesting that BIP1 controls cellular functions specifically required for infection.

## BIP1 is required for the expression of pathogenesis-related genes in appressoria

Genome-wide analysis of the appressorium transcriptome identified 40 genes down-regulated in the *Δbip1* mutant in comparison to wild-type. Most of these *BIP1*-regulated genes (93%) were over-expressed in mature wild-type appressoria as observed for *BIP1* (**Table 1**). Promoters of these *BIP1*-regulated genes shared a TGACTC GCN4/bZIP-like binding motif that bound *in vitro* to BIP1. Mutations of this motif in the promoter of *MGG_08381.5* (*ORF3* from the *ACE1* cluster) abolished *in vitro* BIP1 binding and its appressorium-specific expression (**Fig 8**). Together, these results suggest that in mature appressoria BIP1 activates the expression of a unique set of genes by binding to a TGACTC motif present in their promoters. The specificity of this regulatory network likely results from the fact that *BIP1* expression is restricted to the appressorium.

 *BIP1*-regulated genes encoded proteins with very different cellular functions, *i.e.* small secreted proteins, enzymes involved in secondary metabolism, cell wall degradative enzymes, or GPCRs. However, a shared feature of these proteins is their putative involvement in the infection process. Cuticle and plant cell wall degrading enzymes from the *BIP1* network included a cutinase, a peptidase and a feruloyl esterase. The cutinase, MGG_11966, already described as specifically expressed during penetration [64], could be involved in the degradation of the plant cuticle during appressorium-mediated penetration. The feruloyl esterase encoding gene MGG_03771 is orthologous to *A. niger* feruloyl esterase encoded by *FaeB*

(**S9 Fig**). This enzyme is involved in plant cell wall weakening through the release of phenolic compounds from esterified arabinoxylans and the breakage of diferulic acid linkages between xylans and between xylan and lignin [45]. Therefore, MGG_03771 feruloyl esterase could weaken the plant cell wall to facilitate the penetration of the fungus into the host leaves. Candidate protein effectors (small-secreted proteins, **Table 1**) could also play an important role in the infection process by targeting plant cellular processes [19,20]. Four of them BAS2, BAS3, AvrPi9 and SLP2 (MGG_09693, MGG_11610, MGG_ 12655, MGG_03464, MGG_03468, respectively) have been previously characterized for their specific expression during early stages of infection [37–40]. However, their function and their plant targets are still unknown, except for BAS3 displaying a plant cell death suppressive activity [40]. Secondary metabolites could have important roles in the infection process [65]. *BIP1* controlled the expression of genes involved in their biosynthesis. Most of them belonged to *ACE1* gene cluster [35], involved in the biosynthesis of a possible cytochalasin derivative [66], likely recognized by the rice *Pi33* resistance gene [67]. Our results show that BIP1 is essential for the appressorium-specific expression of genes from the *ACE1* cluster [44]. Finally, *BIP1*-regulated genes encoded five signaling proteins (**Table 1**). Three are members of the PTH11 G-protein-coupled receptor (GPCR) family [68,69], and one is a cAMP GPCR. The biological roles of these GPCRs are unknown. However, they could be involved in the perception of host signals during appressorium-mediated penetration and the induction of signaling pathways required for invasive infectious growth. The contribution of each BIP1-regulated gene to pathogenicity is mostly unknown. Deletion of *ACE1/SYN2* and the effectors *BAS2*, *BAS3*, *SLP2* or *AvrPi9* did not lead to mutants with reduced pathogenicity [36–39]. These results suggest that individually BIP1-regulated genes are not critical for infection. However, the functions they fulfill collectively seem to be crucial for appressorium-mediated penetration and invasive infectious growth.

## Comparison of BIP1 regulatory network to other appressorium regulatory networks

Only few appressorium-specific regulatory networks have been characterized in *M. oryzae*. The best defined are the appressorium-specific gene networks controlled by TFs MST12 and HOX7 activated by the PMK1 MAP kinase signaling pathway [42]. While HOX7 controls the differentiation of appressoria from germ tubes and their maturation, MST12 acts during the formation of the penetration peg. A limited overlap (12 genes) was identified between the genes down-regulated in the *Δmst12* mutant [42] and the BIP1-regulated genes. However, none of these genes was retrieved among the direct target genes of MST12 identified in CHIP seq experiments [42], which suggests that MST12 and BIP1 control different sets of appressorium-specific genes.

The bZIP TF MoAP1 (MobZIP21 in **S4 Table**) is required for infection and other cellular processes such as resistance to oxidative stress and development [31]. The gene network controlled by MoAP1 has been studied by comparing the expression profile of *in vitro*-grown wild-type and mutant mycelia using SAGE [31]. As many as 1180 genes were differentially expressed and the most down-regulated genes were further analyzed for their role in infection (a chitin deacetylase, a C6 TF, a laccase, two proteases, the succinate dehydrogenase MoS-SADH, the acetyltransferase MoACT, and the thioredoxin MoTRX2). Deletion mutants of MoSSADH, MoACT and MoTRX2 mimicked most *Δbzip21* phenotypes including its defects in resistance to oxidative stress and its lack of pathogenicity. Comparison of genes from Mob-ZIP21 and BIP1 regulatory networks showed that they did not overlap, indicating that these regulatory networks control the expression of different sets of genes.

The Zinc-Finger TF MoEITF1 and the bZIP TF MoEITF2 (MobZIP20 in **S4 Table**) are specifically required for infection [33]. Their deletion mutants are reduced in pathogenicity and they do not display defective phenotypes during vegetative growth. Both TFs are only expressed in mature appressoria just before penetration. Analysis of the expression of 30 effector genes in MoEITF1 and MoEITF2 deletion mutants revealed that each of them positively regulates the expression of a small set of effectors during early infection including the avirulence proteins AVR-Pik and AVR-Pizt and the effectors T1REP and T2REP [33]. None of the effectors down regulated in MoEITF1 and MoEITF2 deletion mutants was regulated by BIP1. However, as no genome-wide gene expression analysis was performed for the MoEITF1 and MoEITF2 deletion mutants, the comparison of their networks with the BIP1 network was limited. Still, the BIP1-regulated genes *BAS3* and *AVR-Pi9* were tested for their expression in the MoEITF1 and MoEITF2 deletion mutants. As they were not down-regulated [33], we can assume that the MoEITF1 and MoEITF2 regulatory networks are distinct from the BIP1 network.

For the previously mentioned transcription factor WOR1, whose mutant, like *Δbip1*, is unable to invade the first infected cell, only selected genes were analyzed for altered expression in mutant appressoria. The effector *BAS2*, which was downregulated in *Δbip1*, was overexpressed in *Δwor1*. On the reverse, the effectors *BAS1*, *BAS4*, *AVR-Pita*, *Pwl2* and *MC69*, whose expression were not altered in *Δbip1*, were down-regulated in the WOR1 mutant. Therefore, we assume that the regulatory networks of BIP1 and WOR1 are distinct.

## Conclusion

The discovery of BIP1, which has been missed in previous studies on *M. oryzae* bZIP TFs, sheds new light on the control of fungal gene expression during plant infection. BIP1 is required for the expression of a unique set of genes essential for early invasive infectious growth. This novel appressorium-specific regulatory network opens new perspectives for understanding the control of fungal gene expression during early stages of infection. We show that BIP1 controls the expression of its target genes by binding to a GCN4 bZIP motif present in their promoter and thereby activating their transcription. The specificity of BIP1 seems determined by its expression confined to mature appressoria by transcriptional and post-transcriptional control mechanisms. Additional regulatory networks specific to the appressorium have been characterized in *M. oryzae* (e.g. MST12, MoEITF1, MoEITF2: [33,42]). Their targets genes differ from the one regulated by BIP1, suggesting that these networks act on different sets of genes and processes. Still, each network is not dedicated to a single cellular function. Indeed, genes encoding effectors were found in all these different networks. This observation suggests that different TFs with specific sets of target genes act in parallel to induce the massive transcriptional changes observed at the early stages of infection. BIP1 stands out for its distinctive role in regulating the expression of a specific set of early invasion-related genes within the appressorium. This unique function equips the fungus, even before appressorium-mediated penetration, to adapt to the plant's cellular environment and anticipate its defense responses.

## Materials and methods

### Fungal strains and methods

*M. oryzae* isolate P1.2 used in this study was provided by the Centre de Coopération Internationale en Recherche Agronomique pour le Développement (CIRAD). Isolates, media composition, maintenance, and transformation of the fungus and sexual crosses have been previously described [49, 70]. Strains were cultured on rice medium (rice flour 2% [wt/vol], 0.2% [wt/vol] yeast extract, 1.5% [wt/vol] bacto agar) or on complete medium (CM; 1% [wt/vol] glucose,

0.2% [wt/vol] peptone, 0.1% [wt/vol] yeast extract, 0.1% [wt/vol] Casamino Acids, 0.1% trace elements, and 1X nitrate salts, 1.5% [wt/vol] bacto agar) and incubated at 25°C under 12-h light/dark cycles for 5 to 12 days. For *in vitro* mycelial growth and sporulation rates, all strains were grown on rice medium or CM medium at 25°C. Strains were grown on at least three independent medium plates. After 12 days of growth, conidia were harvested and counted using a tecan. To observe vegetative mycelial growth under stress, 1 mM $H_2O_2$, 0.003% SDS, 0.005% SDS, 400 μg.mL$^{-1}$ calcofluor and 600 μg.mL$^{-1}$ calcofluor were individually added to CM agar medium. A young mycelial plug (5 mm in diameter) of each strain was inoculated in the center of a agar medium containing Petri dish, and the growth rate was assessed by measuring culture diameters after 5, 7 and 9 days of growth.

### REMI insertional mutagenesis

Insertional mutagenesis was performed by the transformation of protoplasts according to the Restriction Enzyme-Mediated Integration (REMI) procedure [34,71], using 1 μg of circular or linearized plasmid pAN7.1 [49] and 0.25 to 10 units of restriction enzyme *Bam*HI, *Bgl*II *Hin*dIII or *Kpn*I. Linearized plasmid for REMI transformation was prepared by digesting pAN7.1 with the restriction enzyme used for REMI transformation, followed by phenol-chloroform extraction and ethanol precipitation.

### Genetic characterization of the M763 insertion mutant

Southern blot analyses of the M763 mutant and co-segregation analyzes in progenies of a cross between M763 and a wild type strain (M4) showed that it contained a single linearized pAN7.1 plasmid insertion. A 0.4 kb *Nde*I-*Ssp*I genomic restriction fragment flanking one of the junctions between the inserted plasmid and the genomic DNA (**Fig 2**) was recovered by plasmid rescue and used to screen a *M. oryzae* genomic cosmid library. A 6 kb *Xho*I-*Bgl*II restriction fragment hybridizing to this probe was identified and subcloned for complementation analysis (**Fig 2**). Pathogenicity was restored in 90% of the transformants of M763 with this restriction fragment, demonstrating that the M763 phenotype is due to the mutation of a gene present within this fragment. A *M. oryzae* cDNA library was screened with the 0.4 kb *Nde*I-*Ssp*I probe, yielding four cDNAs of 2 kb with the same open reading frame as *MGG_08118*.

### Pathogenicity assays, appressorium differentiation, turgor and penetration assays

Screening of the REMI hygromycin-resistant transformants for the loss of pathogenicity was carried out as previously described [49]. The fungal inoculum was prepared by harvesting conidia from 10- to 14-days-old *M. oryzae* rice-agar cultures in sterile water. Pathogenicity assays using droplet inoculation of detached or wounded leaves of barley (*Hordeum vulgare* L.), and spray inoculation of plants of barley and rice (*Oryza sativa*) grown in pots, were performed as previously described [49]. Barley cultivars Express and Plaisant, and rice cultivar Maratelli were used in this study. *M. oryzae* appressorium differentiation and penetration attempts into barley epidermal cells were observed 24, 36, 48 and 72 hai, after staining with lactophenol-Cotton blue and peeling of the epidermal layer as previously described [35]. *M. oryzae* appressorium differentiation and collapsing assays on Teflon membrane was carried out as previously described [49]. Appressorium differentiation was observed on Teflon membrane at 16 hai whereas collapsing rate was assessed at 24 hai by treatments of mature appressoria with PEG8000 at 4% [wt/vol], 10% [wt/vol], and 25% [wt/vol] during 10 minutes. Penetration assays were performed by depositing droplets of 5.10$^4$ conidia.mL$^{-1}$ conidial suspensions either on intact barley leaves or on floating rice leaf sheaths. Penetration assays *in planta* were

observed at 48 hai using an inverted confocal microscope (Objective 63X PLAN APO 1.4 Oil DIC). Fungal material was fixed using ETOH/Chloroforme (3:1), acetic acid (0,15%) overnight and then stained with WGA Alexa 488 fluor at 0.002% during one night. Plant material was stained with calcofluor 0.01% during 15 minutes. Penetration assays on artificial membranes were performed by depositing droplets of conidia ($5.10^4$ conidia.mL$^{-1}$) containing 10 μM hexadecanediol on cellophane membrane (NC2380, thermofisher) previously placed on the top of a 3% (w/v) water–agar layer. The cellophane membranes were observed 48 and 72 hours after inoculation of conidial suspensions by differential interference contrast microscopy (Nikon Eclipse Ni) and z-stacks were realised.

## Nucleic acid methods

Preparation of *M. oryzae* genomic DNA was performed as previously described [72]. Total RNA extractions from conidia or appressoria differentiated on Teflon membrane were performed according to the hot acid phenol method [73]. Cloning, Southern & Northern hybridization, PCR amplification, cosmid and cDNA library screenings were performed according to standard protocols ([74] or manufacturer instructions).

### *BIP1* cloning

For plasmid rescue [75], 1 μg of genomic DNA from *M. oryzae* mutant M763 was digested with *Kpn*I for 5 hours at 37˚C in 20 μL, followed by phenol-chloroform extraction and ethanol precipitation. The pellet was dissolved in ddH$_2$O and incubated with 2 units of T4 DNA ligase (Roche Diagnostics, Mannheim, Germany) in a total volume of 500 μL. After 12 h incubation at 16˚C, the DNA was precipitated with ethanol and dissolved in 5 μL ddH$_2$O. Competent cells of *Escherichia coli* strain DH10B were transformed by electroporation with 2 μL of the ligation product. A 0.4 kb *Nde*I-*Ssp*I genomic restriction fragment of the 11 kb plasmid rescued was used as a probe to screen a genomic cosmid library of *M. oryzae* progeny 96/0/76 [76]. A 6 kb *Xho*I-*Bgl*II cosmid subclone hybridizing to the probe was introduced into plasmid pCB1265 [77] to give pCM763, which was subsequently used for complementation analysis. Transformants of M763 with pCM763 were selected for phosphinothricin resistance on the complex medium defined for this selectable marker [77] and containing 35 mg/L of Bialaphos. The same 0.4 kb *Nde*I-*Ssp*I genomic probe was used to screen a cDNA library made from RNA extracted from a *M. oryzae* mycelium culture grown in a liquid complete medium [49].

### 5' RACE

Transcription initiation sites of *BIP1* were determined by scanning the sequence of 5' RACE products obtained using the GeneRacer kit (Invitrogen), and RNA from conidia or appressoria of *M. oryzae* isolate P1.2 as templates. Reverse transcription was performed with gene-specific RACE1 primer (exon 3 of BIP1 gene, **S5 Table**). Reverse primer RACE2 (exon 2 of *BIP1*) and the RNA primer provided with the kit were used for the PCR amplification of the cDNA. 5' RACE products were cloned into pCR-4Blunt-TOPO.

## Deletion of *BIP1* by targeted gene replacement

The upstream (LB-BIP1; 1.2 kb) and downstream (RB-BIP1; 1.36 kb) regions flanking *BIP1* were obtained by PCR amplification using P1.2 genomic DNA as a template, *Pfu* turbo polymerase (Stratagene, La Jolla, CA) and the couples of primers KO1/KO2-*Sfi*Ia and KO3-*Sfi*Ib/KO4, respectively (**S1 Fig** and **S5 Table**). Primers KO2-*Sfi*Ia and KO3-*Sfi*Ib bear an asymmetric *Sfi*Ia or *Sfi*Ib restriction site. The resulting amplification products were digested with *Sfi*I

(2h, 50˚C). In parallel, the 1.4 kb hygromycin resistance cassette (*hph*) driven by the *Trp*C promoter from *Aspergillus nidulans* was obtained by digestion of plasmid pFV8 (gift from Dr. F. Villalba, Bayer CropScience) with *Sfi*I. The three asymmetric restriction fragments were ligated using T4 DNA ligase to assemble the 3.7 kb pRBIP1 deletion cassette, which was cloned into pCR-4Blunt-TOPO (Invitrogen, Carlsbad, CA). The deletion cassette was amplified by PCR using KO5 and KO6 primers and *Pfu* turbo polymerase. Transformations of P1.2 protoplasts were performed using 3 μg of deletion cassette.

### *BIP1* transcriptional and translational fusions expression vectors

For the transcriptional fusion of the *BIP1* promoter (1345 bp) with 3xeGFP, a restriction fragment containing the *BIP1* promoter (946 bp) and *BIP1* ORF was purified after digestion of pCM763 with *EcoR*I and introduced into plasmid pUC19 (accession number: L09137), to give pUC19-pBIP1-*BIP1*ORF. *BIP1* ORF was removed from pUC19-pBIP1-*BIP1*ORF by digestion with *Nco*I and *Nae*I. It was replaced by a restriction fragment containing 3xeGFP, obtained by digestion of plasmid pUMA647 [78] with the same enzymes, to give pUC19-p*BIP1*-3xeGFP. A p*BIP1*-3xeGFP restriction fragment was purified after digestion of pUC19-p*BIP1*-3xeGFP with *EcoR*I and then reintroduced into the pCM763 vector backbone left over after initial restriction with *EcoR*I. The resulting plasmid was named pCB1265-p*BIP1*-3xeGFP. For the translational fusion, 3xeGFP was fused to the C-terminus of BIP1 under the control of the *BIP1* promoter and terminator (*BIP1*::3xeGFP fusion). A restriction fragment containing 3xeGFP was released from pUMA647 by digestion with *Nco*I and *Nae*I. Its *Nco*I cohesive end was blunted using End-It DNA End-Repair Kit (EPICENTRE Biotechnologies, Madison, Wisconsin, USA). The resulting fragment was introduced into pUC19-pBIP1-*BIP1*ORF after restriction with *Nae*I to give pUC19-pBIP1-*BIP1*ORF-3xeGFP. This plasmid was digested with *EcoR*I and the pBIP1-*BIP1*ORF-3xeGFP restriction fragment was reintroduced into the pCM763 vector backbone, to give plasmid pCB1265-p*BIP1*- *BIP1*ORF-3xeGFP. Transformations of mutant *Δbip1* protoplasts were performed using 3 μg of pCB1265-p*BIP1*-3xeGFP or pCB1265-p*BIP1*-*BIP1*ORF-3xeGFP plasmid previously linearized by *Sca*I. Transformants were selected for phosphinothricin resistance.

### Phenotypic assays of *M. oryzae* GFP transformants

Observations were carried out using detached barley leaves (cultivar Plaisant) infected with droplets of conidial suspensions. eGFP fluorescence was monitored 20–24 hai with a Zeiss epifluorescence microscope equipped with a 488/DM510-550 filter. Nuclei were stained using 4,6-diamidino-2-phenylindole (DAPI, 32 670, Fluka) at 0.8 μg.mL$^{-1}$. Fungal cell wall components were stained with 10 μg/mL of Calcofluor-white (Sigma-Aldrich). The subcellular localization of BIP1-3xeGFP fusion protein was visualized 0–72 hai on Teflon membrane or barley epidermis using a Zeiss LSM510 META inverted confocal microscope, equipped with a 30 mW argon laser, a X63 Plan-Apochromat oil-immersion objective set up at a numerical aperture of 1.4, and Zeiss filter sets 09 (450 to 490 nm band-pass excitation and 515 nm long-pass emission for GFP) and 01 (365/12 nm band-pass excitation filter and 397 nm long-pass emission for Calcofluor and DAPI).

### DNA Microarray analysis

Microarray assays were performed using the Agilent *M. oryzae* oligonucleotide microarray (version 1) platform. 2 μg of total RNA were used to generate fluorescently labeled aRNA probes with the MessageAmp aRNA Amplification kit (Ambion->Agilent) and Cy3 or Cy5 mono-reactive dye (Amersham) as directed in the Ambion protocol. All RNA was quantified

using a NanoDrop ND-1000 spectrophotometer (NanoDrop Technologies) and assayed qualitatively using an Agilent 2100 BioAnalyzer both before and after amplification. Hybridizations were performed as described by Agilent. For each of the three biological replicates, two technical repeats consisting of two slides each were performed. To account for variation due to differences between the two dyes, the labeling dyes for each strain-specific probe were swapped on one slide of each of the technical pairs. Hybridized slides were scanned using an Affymetrix 428 scanner and the resulting images were analyzed using GenePix 4.0 (Axon). Raw expression data were imported into GeneSpring 7.3 (Agilent) and subjected to Lowess normalization, using 20% of the data to fit the Lowess curve at each point in the plot of log intensity versus log ratio. Expression ratio cutoffs of 2.0 and 0.6 were applied to select up-regulated and down-regulated genes, respectively. The raw data generated by the Axon Genepix software was also analyzed using the GeneData Expressionist Refiner and Analyst software package. Data from the 3 biological replicates were imported into GeneData Expressionist Refiner and submitted to a Bayesian background subtraction with a background standard deviation set to 1 and a decay rate of 1000. This method allows the subtraction of background while avoiding a negative signal. Background subtracted data underwent a LOWESS normalization using 10% of the data to fit the smoothing curve. Normalized data was imported into GeneData Expressionist Analyst. For each chip the control and signal channels were kept separate, generating 24 independent data points for each gene (12 wild-type and 12 mutant). An additional median normalization was applied to the whole data set to account for variation between slides. N-way ANOVA was used to determine the effect caused by the mutation on gene expression. A p-value was calculated for each gene, representing the significance that a gene's expression is affected by the mutation. Setting this p-value at $10^{-9}$ yielded 59 genes down-regulated in the mutant as compared to the WT. The intersection of the 59 genes identified by ANOVA analysis and the 81 genes from the Genespring analysis resulted in a gene list of 42 genes with high confidence of being down regulated because of the *BIP1* mutation (**Table 1**). Microarray data have been submitted to the NCBI under accession no. GSE18823.

## Bioinformatic analyses

bZIP transcription factors were identified in six ascomycetes genomes (*Magnaporthe oryzae*, *Neurospora crassa*, *Fusarium graminearum*, *Botrytis cinerea*, *Aspergillus nidulans*, *Phaeosphaeria nodorum*) from Ensembl fungi by detecting the presence of proteins with a bZIP domain using the following database motifs (cd14688, IPR004827, SSF57959, PTHR11462). Amino acid sequences encoding all bZIP domains in these six species of ascomycetes were aligned using Clustal Omega (**S1 Data**). This alignment was visualized using Jalview and was used for building the phylogenetic tree. Phylogenetic analysis was carried out using NGPhylogeny by Maximum likelihood-based inference of phylogenetic trees with Smart Model Selection. The branches were measured using Shimodaira–Hasegawa like Approximate likelihood-ratio test (SH-like *aLRT*). The tree was mid-point rooted and visualized using the Interactive Tree of Life (iTol) tool. Analyses of promoter sequences for potential conserved cis-regulatory elements were carried out using the Weeder algorithm v 1.3 [79]. The algorithm Shuffleseq was used for the randomization of promoter sequences. The library-based algorithm CLOVER [80] was used to identify the TGACTC core motif in the promoters of BIP1 target genes. The algorithm WebLogo (http://weblogo.berkeley.edu/logo.cgi) was used to highlight a consensus sequence for the BIP1 binding site.

## qRT-PCR experiments

RNA was extracted from detached leaves of barley cultivar Plaisant or rice cultivar Sariceltik inoculated with droplets of conidia of either isolate P1.2 or *Δbip1* mutant 24 hai and 17 hai,

respectively. Three independent replicate samples were harvested for each treatment and RNAs were extracted separately. Genomic DNA was removed using DNA-*free* (Ambion). Five μg of total RNA were reverse transcribed using the ThermoScript RT-PCR system (Invitrogen) according to the manufacturer's instructions. The resulting cDNAs were diluted 10 times for analysis. Real-time PCR was carried out with the LightCycler Faststart DNA Master SYBR Green I kit (Roche Diagnostics) using a Light Cycler 1.2 (Roche Diagnostics). Primers used for qPCR are listed in **S5 Table**. Constitutively expressed *EF1-alpha and ILV5* genes were used for normalization as previously described [36]. Results were analyzed using the $2^{-\Delta\Delta Ct} = 2^{(Ctgene\ X\ mutant–Ctgene\ ref\ mutant)–(Ctgene\ X\ wild\ type–Ctgene\ ref\ wild\ type)}$ method [81].

## DNA Binding Assays

Recombinant BIP1 protein was generated *in vitro* using the TnT Quick Coupled Transcription/Translation system (Promega) according to the manufacturer's protocol. The T7 expression construct was amplified from p763c1 using the primers BIP1-RL5, and BIP1-RL-Strep2, gel purified and used directly in the *in vitro* transcription/translation reaction. Double-stranded oligonucleotide probes were constructed by denaturing (95˚C, 10 min) and annealing (room temperature, 30 min) complementary single-stranded oligonucleotides in equimolar amounts to give a final concentration of 6 pmol/μL. Probes (3 pmol) (**S3 Table**) were labeled using T4 polynucleotide kinase (NEB) and $\gamma\ ^{32}$P-ATP, and purified by gel filtration (DTR columns, Edge Biosystems). Binding reactions including 2.5 μL binding buffer (50 mM Tris, pH 7.5, 250 mM NaCl, 2.5 mM dithiothreitol, 2.5 mM EDTA, 5 mM MgCl$_2$, 20% glycerol v/v), 2.5 μL 1 μg/μL poly dI-C, 2.0 μL labeled probe, 2.5 μL BIP1 or control reticulocyte lysate, 9 μl dH2O) were incubated for 20 min at room temperature followed by 15 min on ice before the addition of 5 μL loading buffer and loading on 6% non-denaturing acrylamide gels.

## *MGG_08381.7* transcriptional fusion expression vector

*MGG_08381.7* terminator (1589bp) was amplified with primers *TERMMGG_08381.7NotI+* and *TERMMGG_08381.7EcoRI-* using *Pfu* Turbo polymerase (**S5 Table**). The PCR product, digested by *Not*I and *Eco*RI, was introduced into plasmid pCB1635, which carries a glufosinate resistance marker [77], resulting in plasmid pCB1635-MGG_08381.5. Wild type *MGG_08381.7* promoter (378bp upstream of ATG) and a mutated version whose putative BIP1 binding site (tcgagtcatgctcctagtcatg) was modified (ttggcgtaaccctgtagccatt), were synthesized by GenScript Corp. (Piscataway, USA). A *Not*I restriction site was inserted at the 5' end and a *Nco*I site at the ATG of each promoter. Synthesized promoters were cloned in pUC57 plasmid (accession number: Y14837). Reporter gene *GFP* was obtained by *Nco*I and *Sna*BI digestion of plasmid prom*ACE1:eGFP* [44], and the resulting 1484 bp restriction fragment was inserted in both plasmids pUC57-MGG_08381.7(p) and pUC57-MGG_08381.7(p-mut) digested by *Nco*I and *Sma*I. *MGG_08381.7* promoters (native and mutated) fused to *GFP* were retrieved by *Not*I digestion and introduced in pCB16335-MGG_08381.7 (term) previously linearized by *Not*I. The resulting plasmids were named pMGG_08381.7 (p)-reporter and pMGG_08381.7 (p-mut)-reporter.

## Supporting information

**S1 Fig. Gene replacement of *BIP1*.** A. *M. oryzae BIP1* locus regions used to construct the gene replacement vector. 1.2 kb and 1.36 kb genomic regions (respectively Left Border and Right Border grey boxes) flanking the *BIP1* ORF were amplified using P1.2 genomic DNA and primers shown as arrows (**S5 Table**). The four exons of *BIP1* are shown as black boxes separated by introns. B. Structure of the *BIP1* locus in the *Δbip1* mutants. Hatched boxes correspond to the

hygromycin resistance cassette. Grey boxes represent the Left and Right Border sequences flanking the *BIP1* ORF used to construct the gene replacement vector. C. Analysis of the transformants by Southern blot. Genomic DNA was digested with *Hind*III and probed with the 1.36 kb RB fragment (top) and 0.85 kb *hph* cassette (bottom). Lanes 1, 2, and 3, *Δbip1* transformants; lane 4, Wild type (P1.2).
(PDF)

**S2 Fig. Appressorium differentiation and collapsing rates of strains P1.2, *Δbip1* and *Δbip1*::*BIP1* for different PEG8000 concentrations.** A. Differentiation of appressoria was observed at 16 hai on Teflon membrane. Error bars represent standard deviations. B. Collapsing of appressoria formed on Teflon membrane was assessed 24 hai with PEG8000 at 4%, 10% and 25%. Tree independent experiments with each three different replicate samples were performed. No significant differences in collapsing rates were observed between the three strains (Anova: F = 0.50, P = 0.74, Df = 4).
(PDF)

**S3 Fig. Conidiation rates of strains P1.2, *Δbip1* and *Δbip1*::*BIP1*.** Conidiation rates (conidia. mL$^{-1}$) after 12 days of rice medium cultures. No significant differences in sporulation rates were observed between *Δbip1* and wild-type P1.2 (t-Test day 5, p = 0.84) or between *Δbip1* and *Δbip1*::*BIP1* complemented strain (t-Test day 5, p = 0.87).
(PDF)

**S4 Fig. Mycelial growth of strains P1.2, *Δbip1* and *Δbip1*::*BIP1*.** A. Five-day-old rice medium cultures of P1.2, *Δbip1* and *Δbip1*::*BIP1*. B. Mycelial growth diameters (cm) of 5, 7 and 9-day-old rice medium cultures. No significant difference was observed between *Δbip1* and wild-type P1.2 (t-Test day 5, p = 1) or between *Δbip1* and *Δbip1*::*BIP1* complemented strain (t-Test day5, p = 1). B shows results from three independent experiments each performed with three different replicate samples. Error bars are standard deviations.
(PDF)

**S5 Fig. Stress sensitivity of P1.2 (WT), *Δbip1*, *Δbip1*::*BIP1* strains.** P1.2, *Δbip1*, *Δbip1*::*BIP1* strains were cultured on CM medium without or with stress agent (cell wall integrity stressors: 0.003 or 0.005% SDS (A,B), 200 or 400 µg.mL$^{-1}$ calcofluor (C,D); oxidative stress inducer: 1mM H$_2$O$_2$ (E,F). Colony diameters of strains were measured 5 days after inoculation for cell wall stress (SDS, calcofluor) and 7 days after inoculation for oxidative stress (H$_2$O$_2$). Growth inhibitions were calculated with following formulae (Inhibition rate = (the diameter of untreated strain—the diameter of treated strain)/(the diameter of untreated strain) X 100%). Three independent replicates with three samples per replicate were performed (B,D,F). Error bars are standard deviations. No significant difference was observed between *Δbip1* and P1.2, as well as *Δbip1*::*BIP1* (T-test p-values > 0.05).
(PDF)

**S6 Fig. Observation of appressorium-mediated penetration in rice sheath cells of strains P1.2 and *Δbip1* at 48 hai using confocal microscopy (63X).** No infection hyphae were observed in epidermal cells of rice sheaths infected with *Δbip1* mutant whereas epidermal cells of rice sheaths infected with P1.2 were filled with infection hyphae resulting from penetration events. Fluorescence of WGA-Alexa488-stained fungal cells was excited with 488 nm light and is shown in green. Ap: appressorium, Co: conidium, IH: invasive hyphae, size bar = 10µm.
(PDF)

**S7 Fig. Pathogenicity assays on barley leaves for strains P1.2, *Δbip1* and *Δbip1*::*BIP1*.** Droplets of conidial suspensions (5.10$^4$ conidia.mL$^{-1}$) were deposited on detached intact or

wounded barley leaves or detached wounded rice leaves. Photos were taken 5 dai.
(PDF)

**S8 Fig. bZIP domain of BIP1.** A. Alignment of bZIP domains from BIP1, BIP1 orthologues from *N. crassa* (NcBIP1, NCU03847), *F. graminearum* (FgBIP1, FGRAMPH1_01G06311), *B. cinerea* (BcBIP1, Bcin09g05210), *A. nidulans* (AnBIP1, ANIA_00825) and *P. nodorum* (SNOG_11592). *S. cerevisiae* Gcn4 (ScGCN4, YEL009C) and *S. cerevisiae* Yap1 (ScYAP1, YML007W) TFs were added for comparison. bZIP domains were extracted from protein sequences and aligned using Clustal omega. 100% identical amino acids are highlighted in black. 90–80% similar amino acids are highlighted in dark grey. 70–60% similar amino acids are highlighted in light grey. NLS: nuclear localization signal predicted using Hidden Markov Model for nuclear localization signal prediction. B. Functional domains identified in BIP1 protein using CDD database (bZIP-YAP, CD14688 domain) and previous analysis using YAP1 fungal TFs (A).
(PDF)

**S9 Fig. Phylogenetic analysis of *M. oryzae* feruloyl esterase MGG_03771.** Minimal Evolution tree of selected fungal sequences encoding feruloyl esterases or tannases. *A. niger* FaeA and two orthologous sequences were used as an outgroup. The scale bar shows a distance equivalent to 0.5 amino acid substitutions per site. Bootstrap values (1000 bootstraps) are presented at the nodes. Biochemically characterized proteins are in bold.
(PDF)

**S1 Table. *ACE1* cluster expression in *Δbip1* during barley infection 24 hai.**
(PDF)

**S2 Table. Expression of 15 BIP1-regulated genes during initial stages of rice infection (16hpi).**
(PDF)

**S3 Table. Summary of BIP1 binding to target promoter used for EMSA.**
(PDF)

**S4 Table. Summary of bZIP transcription factors in *M. oryzae*.**
(PDF)

**S5 Table. Primers used in this study.**
(PDF)

**S1 Data. Alignment of bZIP domain sequences used to generate Fig 4 (Fasta format).**
(FASTA)

## Author Contributions

**Conceptualization:** Karine Lambou, Andrew Tag, Jérôme Collemare, Pierre-Henri Clergeot, Jean-Benoit Morel, Roland Beffa, Thomas Kroj, Terry Thomas, Marc-Henri Lebrun.

**Data curation:** Andrew Tag, Philippe Perret.

**Formal analysis:** Karine Lambou, Andrew Tag, Alexandre Lassagne, Jérôme Collemare, Pierre-Henri Clergeot, Philippe Perret, Didier Tharreau, Ronald P. De Vries, Judith Hirsch, Jean-Benoit Morel, Terry Thomas, Marc-Henri Lebrun.

**Funding acquisition:** Jean-Benoit Morel, Roland Beffa, Terry Thomas, Marc-Henri Lebrun.

**Investigation:** Karine Lambou, Andrew Tag, Alexandre Lassagne, Jérôme Collemare, Pierre-Henri Clergeot, Crystel Barbisan, Philippe Perret, Didier Tharreau, Joelle Millazo, Elia Chartier, Judith Hirsch.

**Project administration:** Roland Beffa, Marc-Henri Lebrun.

**Resources:** Jean-Benoit Morel, Roland Beffa, Terry Thomas, Marc-Henri Lebrun.

**Software:** Andrew Tag.

**Supervision:** Karine Lambou, Didier Tharreau, Jean-Benoit Morel, Roland Beffa, Terry Thomas, Marc-Henri Lebrun.

**Visualization:** Karine Lambou, Andrew Tag, Alexandre Lassagne, Jérôme Collemare, Pierre-Henri Clergeot, Crystel Barbisan, Philippe Perret, Ronald P. De Vries, Judith Hirsch.

**Writing – original draft:** Karine Lambou, Andrew Tag, Jérôme Collemare, Pierre-Henri Clergeot, Jean-Benoit Morel, Roland Beffa, Thomas Kroj, Terry Thomas, Marc-Henri Lebrun.

**Writing – review & editing:** Ronald P. De Vries.

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
