## [Decision Letter · Decision Letter 0]

24 Apr 2023

Dear Lecturer Lambou,

Thank you very much for submitting your manuscript "The bZIP transcription factor BIP1 of the rice blast fungus is essential for infection and regulates a specific set of appressorium genes" for consideration at PLOS Pathogens. As with all papers reviewed by the journal, your manuscript was reviewed by members of the editorial board and by several independent reviewers. In light of the reviews (below this email), we would like to invite the resubmission of a significantly-revised version that takes into account the reviewers' comments.

This work focused on characterizing a new bZIP transcription factor, *BIP1*, found to be required for appressorial function and the expression of a subset of genes including those encoding effectors, cell wall modifying enzymes or enzymes required for secondary metabolism that might (or in some cases have been shown previously) contribute to virulence. In the absence of *BIP1*, the mutant forms appressoria, but no invasive hyphae are detected, suggesting penetration is impaired. The work is interesting and might shed new light on the genetic regulation of infection.

The reviewers suggest how the paper might be improved, but generally hit on one of two major flaws of the paper that I am also concerned about, namely that it is not shown what the physiological basis for infection impairment is and, I might also add, it is not shown what underlying gene expression changes in Δ*bip1* directly result in the loss of appressorium penetration and/ or IH development. Thus, the work currently lacks critical details about Bip1 function that need to be resolved.

With regards to appressorial physiology, it must be determined if loss of infection is due to impaired turgor or other maturation defects as suggested by Reviewers 1 and 3 or whether, as Reviewer 2 indicates, penetration pegs are formed and capable of penetrating host rice cells. I might also add that the authors should check whether Δ*bip1* appressoria are defective in mucilage for adhesion. A recent study of a mutant, Δ*sps1*, which is also Δ*bip*1-like but was not included in the list on line 517, showed it produced a peg but was reduced for adhesion, which impaired turgor, as determined using cellophane assays and turgor measurements (Rocha et al. 2020; DOI: 10.1038/s41564-020-0786-x). Thus, the authors should similarly assess Δ*bip1* appressoria on cellophane, as well as measure their turgor, in order to determine if they form pegs or have altered mucilage production that would affect adhesion and turgor. Alternatively, Δ*bip1* might form normal appressoria and pegs which penetrate host cells but fail to elaborate IH. This would be harder to assess, but not impossible by using, for example, SEM to show peg penetration takes place. I agree with Reviewer 2 that rice experiments are essential throughout. Other cellular markers of appressorium formation, as suggested by Reviewer 3, might also be informative here. Editorially, therefore, we request that turgor and penetration peg formation data be added to the paper.

With regards to my second point, I am also concerned that it is not known which Bip1-controlled genes are required for appressoria penetration and/ or IH elaboration. Knocking out in WT some downstream candidate genes to show they result in a similar Δ*bip1*-phenotype, or constitutively expressing one or two candidates in Δ*bip1,* might be useful here.

Other good points are raised by the reviewers, such as whether conidiation is affected in Δ*bip1*, and these should all be addressed.

We cannot make any decision about publication until we have seen the revised manuscript and your response to the reviewers' comments. Your revised manuscript is also likely to be sent to reviewers for further evaluation.

Sincerely,

Richard A Wilson

Academic Editor

PLOS Pathogens

Bart Thomma

Section Editor

PLOS Pathogens

Kasturi Haldar

Editor-in-Chief

PLOS Pathogens

orcid.org/0000-0001-5065-158X

Michael Malim

Editor-in-Chief

PLOS Pathogens

orcid.org/0000-0002-7699-2064

This work focused on characterizing a new bZIP transcription factor, BIP1, found to be required for appressorial function and the expression of a subset of genes including those encoding effectors, cell wall modifying enzymes or enzymes required for secondary metabolism that might (or in some cases have been shown previously) contribute to virulence. In the absence of BIP1, the mutant forms appressoria, but no invasive hyphae are detected, suggesting penetration is impaired. The work is interesting and might shed light on

The reviewers suggest how the paper might be improved, but generally hit on one of two major flaws of the paper that I am also concerned about, namely that it is not shown what the physiological basis for infection impairment is and, I might also add, it is not shown what underlying gene expression changes in Δbip1 directly result in the loss of appressorium penetration and/ or IH development. Thus, the work currently lacks critical details about Bip1 function that need to be resolved.

With regards to appressorial physiology, it must be determined if loss of infection is due to impaired turgor or other maturation defects as suggested by Reviewers 1 and 3 and determine, as Reviewer 2 indicates, whether or not penetration pegs are formed and capable of penetrating host rice cells. I might also add that the authors should check whether Δbip1 appressoria are defective in mucilage for adhesion. A recent study of a mutant, Δsps1, which is also Δbip1-like but was not included in the list on line 517, showed it produced a peg but was reduced for adhesion, which impaired turgor, as determined using cellophane assays and turgor measurements (Rocha et al. 2020; DOI: 10.1038/s41564-020-0786-x). Thus, the authors here should similarly assess Δbip1 appressoria on cellophane, as well as measure their turgor, in order to determine if they form pegs or have altered mucilage production that would affect adhesion and turgor. Alternatively, Δbip1 might form normal appressoria and pegs which penetrate host cells but fail to elaborate IH. This would be much harder to assess, but not impossible by using, for example, SEM to show peg penetration takes place. I agree with Reviewer 2 that rice experiments are essential throughout. Other cellular markers of appressorium formation, as suggested by Reviewer 3, might also be informative here. Editorially, therefore, we request that turgor and penetration peg formation data be added to the paper.

With regards to my second point, I am also concerned that it is not known which Bip1-controlled genes are required for appressoria penetration and/ or IH elaboration. Knocking out in WT some downstream candidate genes to show they result in a similar Bip1-phenotype or constitutively expressing one or two candidates in Δbip1 might be useful here.

Other good points are made by the reviewers, such as whether conidiation is affected in bip1, and these should all be addressed.

Reviewer's Responses to Questions

**Part I - Summary**

Reviewer #1: This manuscript describes the identification and characterization of a bZIP transcription factor BIP1 of the rice blast fungus essential for infection and regulation of appressorium genes. All experiments were straight forward and all data were logically interpreted. However, there are a few things to be clarified to improve the quality of this manuscript.

Reviewer #2: This manuscript details the identification of a transcription factor involved in the regulation of appressorium penetration in Magnaporthe oryzae. The authors report on a novel gene involved in the regulation of appressorium-mediated penetration. This data provides insight into a network of genes that are involved in M. oryzae penetration and how this is crucial for pathogenicity. These findings may provide avenues for the generation of disease mitigation strategies.

This manuscript is very well written. The authors provide a good overview of appressoria structure and of transcription factor networks in M. oryzae. However, this manuscript would benefit from the addition of experiments that would further confirm the findings of the authors.

Reviewer #3: This is a very well-written and interesting manuscript that provides important new insight into the role of a previously overlooked basic leucine zipper (bZIP) transcription factor BIP1, in orchestrating appressorium-mediated plant infection by the blast fungus, Magnaporthe oryzae. While transcriptional control of �bip1 insertion/deletion mutants are able to form melanized appressoria, but these are non-functional and cannot penetrate leaves. The authors demonstrate that BIP1 controls the expression of 40 pathogenicity-related genes, all of which share a GCN4/bZIP-binding DNA motif. These genes include known secreted effectors and enzymes involved in secondary metabolism, among others. The authors convincingly demonstrate that Bip1 binds to these sequences in an in vitro assay. The authors go on to show that mutation of this DNA motif abolished expression of a secondary metabolism gene cluster. Overall, the data are of a high-quality and support the conclusions made by the authors.

Without looking at additional subcellular makers it’s not possible to conclude that appressoria from bip1 mutants are fully mature. As the authors note in Line 517, using light microscopy alone, bip1 mutants look identical to nox2 and mst12 knockout mutants. However, careful cell biological analysis using various fluorescent subcellular markers reveals differences in the terminal phenotypes. It’s probably beyond the scope of this manuscript, but it would be interesting to try and further resolve the nature of the terminal phenotype.

Figure 5. I’m curious as to whether the use of 3xeGFP was necessary due to low or undetectable fluorescence with single copy eGFP, or whether this was simply chosen by default?

Further, was nuclear localization of Bip1-3xeGFP present in all/most nuceli, or was there heterogeneity within the population? It might be nice to have some quantitation, but this is not absolutely necessary. It might also be compelling to perform time-lapse imaging to resolve precisely when Bip1-3xeGFP becomes localized to the appressorium nucleus. How long after mitotic division and nuclear migration is it before fluorescence is detectable in the nucleus? Again, not essential, but if it can be done easily, it might add some value.

Figure 6 and 7. I’m curious as to how many times the DNA-binding assays were performed? If multiple replicates already exist, it might be nice to quantify these.

A very minor point: Lines 360 and 408 mention “randomly” selecting genes/promotors to study further, but I couldn’t see any mention in the methods of how this process was randomized. Did the authors just select their favourites or was this truly randomized, and if so how?

**Part II – Major Issues: Key Experiments Required for Acceptance**

Reviewer #1: 1. Deletion of BIP gene still formed well melanized appressoria, but did not penetrate, even wounded leaves. Instead of just saying defective, authors may check the turgor of appressoria formed by deletion mutant using plasmolysis/cytolysis. Further it would be better authors present all phenotypes of the mutant measured in this study including stress responses to speculate more on defective in invasive growth.

2. Expression of BIP1 gene. There is difference between on the Teflon (conidia and appressoria) and barley leaves. What is the explanation, plant (e.g. how about on the onion?) or host effect? RNAseq was done the samples from Teflon. However, it is not clear what type of RNA sample used for oligonucleotide microarray. Authors identified 42 down regulated genes from bip1 deletion mutant and binding motif TGACTCG. These 42 were from microarray data. How about RNAseq data? Since authors already has RNAseq data of bip1 deletion mutant during appressorium formation, more comprehensive genome-wide identification/analysis would be available.

3. Identification of BIP1 binding motif. It would be better if authors could provide genome-wide presence of this motif, not limited to 42 genes!

Reviewer #2: Comments:

Many experiments conducted in this manuscript are performed in barley. All experiments should be conducted using rice plants to fully validate the results obtained using barley. This is especially important when conducting gene expression profile and microscopy analyses.

Lines 150-151: Referencing Figure 1 and Figure 3 does not support the claims by the authors. Results from infection on wounded leaves must be shown in the manuscript and should be performed using rice.

In line 223, the authors mention BIP1 has a nuclear localization signal. However, this is not represented as mentioned in Supplementary Figure 2. It would be ideal if the authors included a schematic representation of the sequence of BIP1, highlighting the NLS and other domains within the gene.

In addition to the previous point, the authors can generate truncations of the BIP1 protein, with or without the NLS, to show the functionality of the NLS in BIP1. This would strongly support the findings from the authors in Figure 5D that indeed BIP1 localizes to the nucleus of the appressoria via its operational NLS.

In the materials and methods on line 784, the authors state that they use onion epidermis for experiments. However, there are no figures depicting the results of using onion for microscopy work. The authors should include the results from these experiments that are listed in this section.

While the authors state that there is no additional phenotype seen in this BIP1 mutant, the authors need to include morphological characterization in axenic conditions, conidiation data, and appressorium formation assays using the BIP1 mutant and WT strain.

This manuscript would benefit from conducting cell wall integrity and oxidative stress tests on the BIP1 mutant strain compared to the WT to see if there are additional defects with the mutant strain. Even if these findings show negative data, the authors can include these results in the supplementary section to depict the full profile of this fungal mutant strain.

How can the authors guarantee that the strain is not penetrating rather than not proliferating inside the cell? Figure 1A shows small lesions when infecting barley with the BIP1 mutant. If this mutant is highly involved in penetration peg formation, the authors should present penetration rate statistics on rice between the WT and BIP1 mutant strain to show the overall percentage of infection.

Additional experiments need to be conducted to show that the penetration peg is defective in the BIP1 mutant. Microscopy work using rice and the BIP1 mutant strain at varying time points of infection would allow readers to see defects in penetration peg formation and invasive hyphae growth.

How is the BIP1 localization in the conidia versus the mycelia in the 3xGFP strain? This data would complement the gene expression profile presented in Figure 5A.

Authors should consider statistics analysis for the pathogenicity assays and appressoria formation.

Reviewer #3: n/a

**Part III – Minor Issues: Editorial and Data Presentation Modifications**

Reviewer #1: 1. Promotor mutation of MGG08381. Are there any phenotypes including appressorium formation and invasive growth?

2. In Figure 5, what are the units of expression levels (A and B)? Sp should be Conidia?

Reviewer #2: (No Response)

Reviewer #3: (No Response)

PLOS authors have the option to publish the peer review history of their article (what does this mean?). If published, this will include your full peer review and any attached files.

Reviewer #1: No

Reviewer #2: No

Reviewer #3: No
---

## [Decision Letter · Decision Letter 1]

11 Sep 2023

Dear Lecturer Lambou,

Thank you very much for submitting your manuscript "The bZIP transcription factor BIP1 of the rice blast fungus is essential for infection and regulates a specific set of appressorium genes" for consideration at PLOS Pathogens. As with all papers reviewed by the journal, your manuscript was reviewed by members of the editorial board and by several independent reviewers. In light of the reviews (below this email), we would like to invite the resubmission of a significantly-revised version that takes into account the reviewers' comments.

Because penetration of bip1 appressoria were not assayed in the revised version of the paper as requested, it is not possible to conclude that this mutant is defective in appressorial penetration, thus the conclusions of the work are unsafe. Cellophane assays of peg formation are standard means of addressing this. In the absence of determining whether or not  bip1 appressoria form penetration pegs, an alternative explanation is that this mutant does form pegs that penetrate leaf surfaces, but Bip1 is then required for elaborating primary hyphae from the peg, or for elaborating invasive hyphae from primary hyphae. Considering pegs and primary hyphae cannot be observed in planta by confocal microscopy, the role of Bip1 in infection-related development is currently ambiguous. To reach the high bar for publication in PLoS Pathogens, bip1 penetration peg formation must be assayed. 

We cannot make any decision about publication until we have seen the revised manuscript and your response to the reviewers' comments. Your revised manuscript is also likely to be sent to reviewers for further evaluation.

Sincerely,

Richard A Wilson

Academic Editor

PLOS Pathogens

Bart Thomma

Section Editor

PLOS Pathogens

Kasturi Haldar

Editor-in-Chief

PLOS Pathogens

orcid.org/0000-0001-5065-158X

Michael Malim

Editor-in-Chief

PLOS Pathogens

orcid.org/0000-0002-7699-2064

Reviewer's Responses to Questions

**Part I - Summary**

Reviewer #2: This manuscript details the identification of a transcription factor involved in the regulation of appressorium penetration in Magnaporthe oryzae. The authors report on a novel gene involved in the regulation of appressorium-mediated penetration. This data provides insight into a network of genes that are involved in M. oryzae penetration and how this is crucial for pathogenicity. These findings may provide avenues for the generation of disease mitigation strategies.

The authors have included many of the comments as suggested by the Reviewers. Specifically, the experiments conducted have been done in rice plants to agree with the stated findings. Some additional pieces of information and experiments are needed, however, for clarity and validation of results.

Reviewer #3: I appreciate the authors' inclusion of new data from a cytorrhysis assay, which helps to further resolve the phenotype of the Bip1 mutant in demonstrating that turgor production is comparable with the wild type, which strengthens the manuscript. However, without further cell biological investigation of appressorium subcellular architecture, using existing fluorescently-tagged fusion constructs, I would still err on the side of caution when referring to the extent of appressorium "maturation" in Bip1 mutants. I don't think that the ability to generate turgor pressure alone, confirms that the appressoria are necessarily fully matured in other respects (e.g cytoskeleton remodelling), although they might well be. Given that the authors outline the importance of the septin and actin cytoskeleton in their intro Line 90 "ROS production by NADPH oxidases as well as septin GTPase-dependent actin organization of the cytoskeleton are required for the cell re-polarization at the pore and the formation of the penetration peg" it would have been nice to examine these elements directly, given the nature of the Bip1 mutant phenotype.

**Part II – Major Issues: Key Experiments Required for Acceptance**

Reviewer #2: Line 153-154: This statement needs to be proven on the cellophane membrane in order to conclude that penetration is not happening due to an impairment in penetration peg formation. Since the authors were unable to determine if a penetration peg was developed or not, it is not appropriate to state that the Bip1 mutant ‘presumably did not produce a penetration peg’ given that it may be produced, but be defective in generating invasive hyphae.

It seems that a small amount of lesions are seen when infecting barley with the Bip1 mutant strain. How can the authors justify the small lesions seen in the mutant if no penetration peg was present? Authors need to be cautious about saying that the penetration peg was arrested in the mutant.

Reviewer #3: n/a

**Part III – Minor Issues: Editorial and Data Presentation Modifications**

Reviewer #2: It would be ideal to perform a time course of infection for a prolonged time using the Bip1 mutant to see if there is a delay in infection, both in visual disease symptoms and microscopy. If nothing is seen, the manuscript should at least reflect this finding.

Plant growth and infection conditions should be specified in the materials and methods.

Reviewer #3: n/a

PLOS authors have the option to publish the peer review history of their article (what does this mean?). If published, this will include your full peer review and any attached files.

Reviewer #2: No

Reviewer #3: No
---

## [Editor Report · Decision Letter 2]

14 Dec 2023

Dear Lecturer Lambou,

Thank you very much for submitting your manuscript "The bZIP transcription factor BIP1 of the rice blast fungus is essential for infection and regulates a specific set of appressorium genes" for consideration at PLOS Pathogens.  We are likely to accept this manuscript for publication, providing that you modify the manuscript according to my recommendations, specifically:

The inclusion of the cellophane penetration assay has strengthened the conclusions of the paper, but I have three further comments raised by the new version of the paper:

1. Bip1 is shown to not be required for penetration peg formation or, according to the cellophane assay, penetration, but is required for establishing IH in the first infected cell. This is stated in the discussion, but in the abstract and elsewhere, there are statements such as "BIP1 is essential for appressorium-mediated penetration into plant leaves but not into artificial cellophane membranes", which is misleading, as it suggests a role for Bip1 in penetration on leaves. Please make clear throughout that on the basis of the results, Bip1 is not likely required for penetration but is required for establishing early biotrophic growth in host cells. 

2. Impaired penetration is shown for barley (Fig 3) and rice (S6 Fig). Please note that the figure legend for S6 Fig has been swapped with that for S4Fig, please correct. Also, many readers are interested in rice blast rather than barley blast. Therefore, it is important that the confocal images in S6Fig, stained with WGA, be moved to the main figure alongside the barley results. However, these image panels need to be improved. Both a DIC or brightfield panel and a merged panel should be included, and the appressoria on the surface and points of IH spread to neighbouring cells should be indicted. The current images are hard to interpret and it looks to me like the stained IH for WT and Bip1 complement are hyphae on the surface of the leaf. It may be necessary to show surface images of the three strains to indicate appressoria, and a lower depth image to show IH produced by the WT and the complement.

3. Because bip1 can penetrate cellophane but cannot infect wounded tissues, the most parsimonious explanation is that Bip1 is required for expressing genes in appressoria that are needed for establishing biotrophic growth, but not for penetration. This important point needs to stressed throughout, as it suggests genes required by IH for colonizing hot cells are "pre-expressed" before penetration, perhaps in order to condition the fungus for the host cell prior to infection. I do not believe this was clearly stated in the text, but it is a main finding of the work and perhaps should be added to the abstract.

Sincerely,

Richard A Wilson

Academic Editor

PLOS Pathogens

Bart Thomma

Section Editor

PLOS Pathogens

Kasturi Haldar

Editor-in-Chief

PLOS Pathogens

orcid.org/0000-0001-5065-158X

Michael Malim

Editor-in-Chief

PLOS Pathogens

orcid.org/0000-0002-7699-2064

Reviewer Comments (if any, and for reference):

Figure Files:

Data Requirements:

Reproducibility:

References:

---

## [Editor Report · Decision Letter 3]

4 Jan 2024

Dear Lecturer Lambou,

We are pleased to inform you that your manuscript 'The bZIP transcription factor BIP1 of the rice blast fungus is essential for infection and regulates a specific set of appressorium genes' has been provisionally accepted for publication in PLOS Pathogens.

In addition, the inclusion of the rice leaf sheath images to fig 3 is good, but it is not immediately obvious that these are from fixed samples. Thus, the images may look atypical to some readers expecting live-cell imaging (for example, the rice cell walls are not visible even in the bright field image, and I assume this is due to the fixation process?). My recommendation is that it is noted in the figure legend that the samples are fixed (ie. "..and fungal staining with WGA-Alexa488 after fixation")

Best regards,

Richard A Wilson

Academic Editor

PLOS Pathogens

Bart Thomma

Section Editor

PLOS Pathogens

Kasturi Haldar

Editor-in-Chief

PLOS Pathogens

orcid.org/0000-0001-5065-158X

Michael Malim

Editor-in-Chief

PLOS Pathogens

orcid.org/0000-0002-7699-2064
---

## [Editor Report · Acceptance letter]

17 Jan 2024

Dear Lecturer Lambou,

We are delighted to inform you that your manuscript, "The bZIP transcription factor BIP1 of the rice blast fungus is essential for infection and regulates a specific set of appressorium genes," has been formally accepted for publication in PLOS Pathogens.

Best regards,

Michael Malim

Editor-in-Chief

PLOS Pathogens

orcid.org/0000-0002-7699-2064